# In vivo patch-clamp recordings reveal distinct subthreshold signatures and threshold dynamics of midbrain dopamine neurons

Kanako Otomo[1,3], Jessica Perkins[2,3], Anand Kulkarni[2], Strahinja Stojanovic[1], Jochen Roeper[1,4✉] & Carlos A. Paladini[2,4✉]

The in vivo firing patterns of ventral midbrain dopamine neurons are controlled by afferent and intrinsic activity to generate sensory cue and prediction error signals that are essential for reward-based learning. Given the absence of in vivo intracellular recordings during the last three decades, the subthreshold membrane potential events that cause changes in dopamine neuron firing patterns remain unknown. To address this, we established in vivo whole-cell recordings and obtained over 100 spontaneously active, immunocytochemically-defined midbrain dopamine neurons in isoflurane-anaesthetized adult mice. We identified a repertoire of subthreshold membrane potential signatures associated with distinct in vivo firing patterns. Dopamine neuron activity in vivo deviated from single-spike pacemaking by phasic increases in firing rate via two qualitatively distinct biophysical mechanisms: 1) a prolonged hyperpolarization preceding *rebound bursts*, accompanied by a hyperpolarizing shift in action potential threshold; and 2) a transient depolarization leading to high-frequency *plateau bursts*, associated with a depolarizing shift in action potential threshold. Our findings define a mechanistic framework for the biophysical implementation of dopamine neuron firing patterns in the intact brain.

[1] Institute of Neurophysiology, Neuroscience Center, Goethe University, Frankfurt, Germany. [2] University of Texas at San Antonio Neurosciences Institute, San Antonio, USA. [3]These authors contributed equally: Kanako Otomo, Jessica Perkins. [4]These authors jointly supervised this work: Jochen Roeper, Carlos A. Paladini. ✉email: roeper@em.uni-frankfurt.de; carlos.paladini@utsa.edu

The midbrain dopamine system is necessary for essential brain functions related to reward-based learning, motivation, action, and cognition[1–4]. Loss or dysregulation of dopamine neuron subpopulations located in the substantia nigra pars compacta (SNc) and ventral tegmental area (VTA) perturbs crucial circuits leading to major brain disorders, including Parkinson Disease (PD)[5], schizophrenia[6], depression[7], and addiction[8]. However, describing the dynamics of subthreshold ionic conductances that cause dopamine neuron spiking activity in vivo has remained a challenge.

Single-unit extracellular recordings from non-human primates and rodents have provided major insights into the computational roles of dopamine neuron firing, most prominently in the context of reward-based learning[4,9,10]. Individual dopamine neurons signal prediction errors by transient, sub-second changes in their firing rates. A recent study also suggested that prediction error coding by dopamine neurons might occur in a distributed manner[11]. Transient cue- or reward-associated increases in firing rate – from a low tonic background firing frequency (in the range of about 1–8 Hz) – are thought to encode positive reward prediction errors (RPEs), while transient reductions or pauses in baseline firing are believed to encode negative RPEs, induced for instance by the omission of an expected reward. Increasingly, experimental evidence points to additional dopamine functions, such as signaling salience, novelty and action control[12,13], as well as aversion[14]. This functional diversity might also be reflected by the emerging molecular[15,16], cellular, and anatomical diversity of the midbrain dopamine system[17–23]. Even within canonical RPE signaling, the interpretation of the transient burst has become more complex. It is currently divided into a short-latency component, which might signal the sensory characteristics of the stimulus (e.g. intensity); and a long-latency component, which might represent the predictive properties of the stimulus[10]. In addition, recent progress has been made in characterizing the underlying arithmetic of single dopamine neuron activity in the context of RPE signaling[24].

To approach in vivo electrophysiology from the perspective of the dopamine neuron, it is reasonable to assume that it integrates excitatory and inhibitory inputs that are derived from heterogeneous afferents[25], thereby dynamically tuning its intrinsic pacemaker activity. This process enables the generation of burst-pause firing patterns, which in turn contribute to temporally resolved extracellular dopamine concentration transients in axonal target regions[26]. However, how inputs are integrated in dopamine neurons in vivo has remained unknown for the last three decades due to the lack of available tools. While the in vivo extracellular recording method does reveal spontaneous firing patterns of dopamine neurons, it is blind to the causal subthreshold processes that drive them. Intracellular recordings are unique in their ability to directly monitor and manipulate membrane potentials with full temporal resolution. Yet, the only available in vivo intracellular data on dopamine neurons came from the pioneering studies by Grace and Bunney[27–31] in the early eighties. Here, we provide the first large dataset on subthreshold membrane potential properties of identified dopamine neurons in vivo by establishing a robust method for deep in vivo patch-clamp recordings in anesthetized adult mice. Our dataset comprises 110 spontaneously active, immunocytochemically identified dopamine neurons, which were mapped to their respective anatomical positions within the SNc and VTA. We discovered two qualitatively distinct subthreshold membrane potential signatures associated with transient high-frequency firing, which we named rebound and plateau bursting. Moreover, we demonstrate that in vivo pacemaking of dopamine neurons exhibits tightly controlled action potential thresholds, while rebound and plateau bursts are characterized by opposing shifts in action potential thresholds. Thus, our study provides a framework for mechanistic studies into in vivo firing patterns of midbrain dopamine neurons.

## Results

**Stable in vivo whole-cell recordings of immunocytochemically identified dopamine neurons reveal subthreshold membrane potential.**

By establishing deep in vivo patch-clamp recordings, we obtained in vivo whole-cell recordings from 110 spontaneously firing, identified midbrain dopamine neurons in isoflurane-anaesthetized, adult C57BL/6 mice ($n = 110$, $N = 78$; Fig. 1). All neurons included in this study were filled with neurobiotin or biocytin and successfully recovered for anatomical mapping and post-hoc neurochemical identification as dopaminergic using tyrosine hydroxylase (TH) immunocytochemistry (Fig. 1c). Our coverage included most subregions of midbrain dopamine regions, with the majority of our recorded neurons located in the rostro-dorsal (i.e. parabrachial) VTA and rostro-medial SNc, with fewer neurons in more lateral aspects of the SNc and caudal ventro-medial (i.e. paranigral) VTA regions (Fig. 1d). Due to the unbiased nature of in vivo patch-clamping, we also encountered TH-negative, electrically silent cells with membrane potentials below $-50$ mV as well as TH-negative, fast-firing cells (>10 Hz) with narrow action potentials (data not shown). Moreover, in addition to the 110 spontaneously active dopamine neurons included in this study, we observed 2 electrically silent TH-positive cells that only fired in response to depolarizing current injections, and 3 other TH-positive neurons that fired spontaneously less than one spike per minute (Supplemental Fig. 1). Under isoflurane-anaesthesia, the technical and biological stability of recordings in the whole-cell configuration were not a major limiting factor (Supplemental Fig. 2), and recordings could last for up to 60 minutes with stable input resistances. However, we usually terminated the recording in a controlled fashion via the outside-out patch configuration to retrieve the cell for post-hoc immunohistochemistry. Our dataset includes dopamine neurons recorded in male ($n = 100$, $N = 73$) and female ($n = 12$, $N = 7$) mice from two different C57BL/6 substrains (C57BL/6N ($n = 92$, $N = 65$) and C57BL/6J ($n = 20$, $N = 15$). We also used two different variants of the internal pipette solution with a small difference in estimated free calcium concentrations of 40 nM ($n = 8$, $N = 6$) and 80 nM ($n = 104$, $N = 74$). As there were no significant differences in the mean firing rates (FR) and firing variabilities (quantified as coefficient of variation (CV) of interspike intervals (ISI)) among these data sets, they were pooled (see Supplemental Table 1). The input resistances were calculated for a subset of these neurons, which had a median of ~300 MΩ (Median (Interquartile range, IQR) = 295.2 (228.1–349.8) MΩ; $n = 42$, $N = 36$; Fig. 1g), an order of magnitude higher than the previously reported values using sharp microelectrodes[31] and more similar to a median of ~450 MΩ for synaptically isolated dopamine neurons recorded in vitro with the same pipette solution (Median (IQR) = 446.6 (301.3–656.7) MΩ; $n = 71$, $N = 11$; see Supplemental Table 2). Assuming that intrinsic conductances are similar when recording either in vitro or in vivo, our results imply a contribution of ~1 nS synaptic and 2 nS intrinsic conductances in dopamine neurons in vivo. This ratio suggests that intrinsic conductances of dopamine neurons are less likely to be shunted in comparison to other types of central neurons[32,33]. In addition, subthreshold membrane potential responses to in vivo hyperpolarizing current injections were recorded in a subset of cells, which revealed a diversity reminiscent of previous in vitro studies (Supplemental Fig. 3)[20].

Figure 1b shows a typical in vivo patch-clamp recording of an identified midbrain dopamine neuron. The recording, when presented on an expanded timescale, displays episodes of

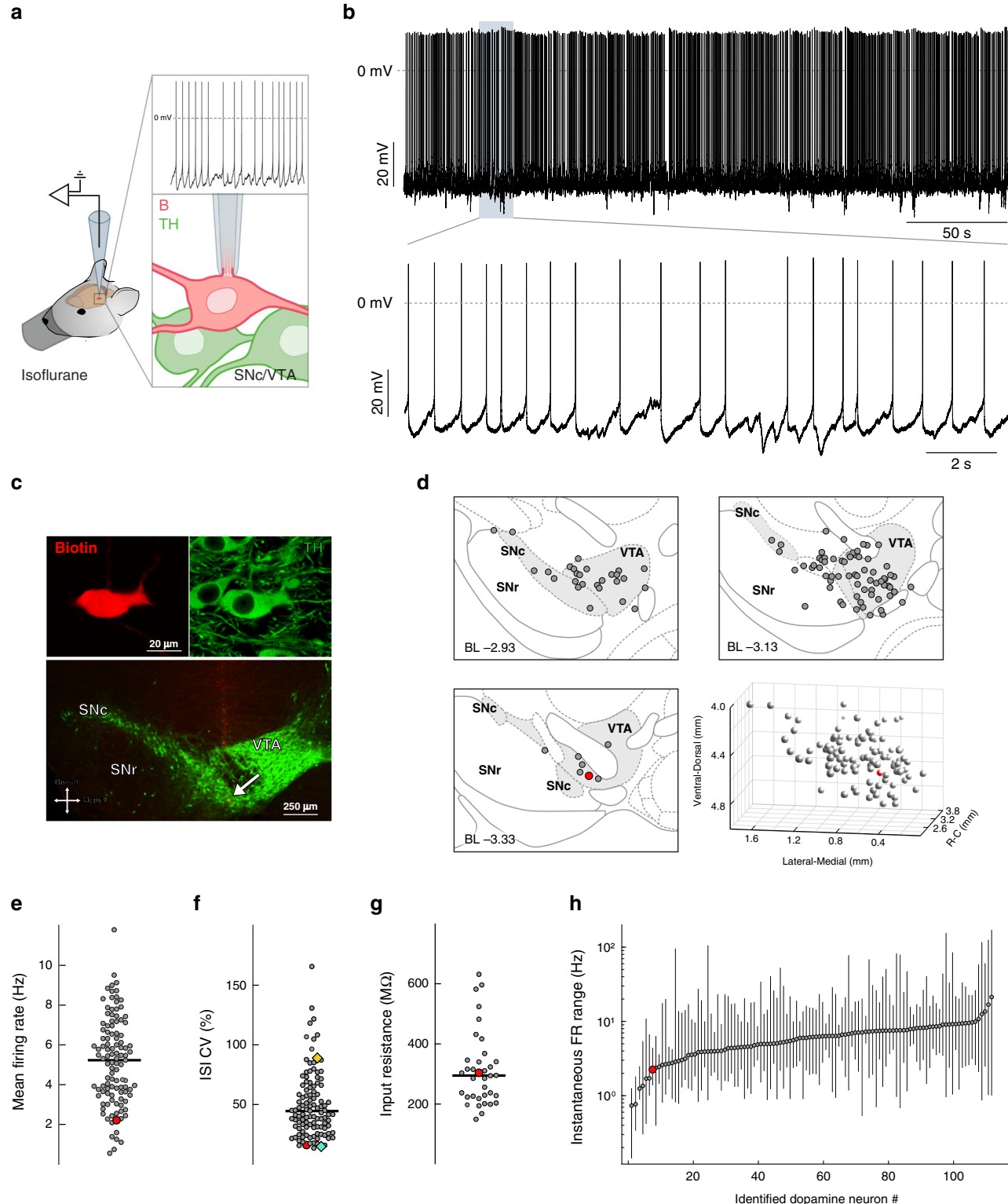

overshooting action potentials interspersed by resting periods where the membrane potential is hyperpolarized below the threshold for action potentials (Fig. 1b, bottom trace). Post-hoc immunocytochemistry identified this cell as a dopamine neuron localized in the ventral-caudal VTA (Fig. 1c, bottom image; and 1d, red circle).

Figure 1e, f show the mean firing rates and CVs of spontaneous firing activity from 110 dopamine neurons, respectively (mean firing rates median (interquartile range, IQR) = 5.25 (3.56–6.76);

CV median (IQR) = 42.49 (30.29–65.34)). Among all measured variables, only mean firing rate was found to be significantly different between SNc and VTA dopamine neurons (Supplemental Fig. 4). In accordance to previous in vivo extracellular studies, we also analyzed the firing patterns of dopamine neurons using the traditional burst firing heuristic established by Grace and Bunney[28]. The results, expressed as the percentage of spikes fired in bursts (%SFB), were again comparable to extracellular recordings using the same anaesthesia[19] (see Supplemental

**Fig. 1 In vivo whole-cell recordings of identified ventral mesencephalon dopamine neurons. a**. Schematic representation of the experimental setup. **b**. Representative recording of spontaneous in vivo electrical activity from a whole-cell-recorded and neurochemically-identified dopamine neuron in the ventral tegmental area (VTA) of an adult C57Bl/6 J mouse under isoflurane anesthesia. The upper trace displays action potentials with overshoot during a stable recording for > 5 minutes. The higher temporal resolution of the electrical activity in the lower trace allows visual identification of subthreshold events such as large hyperpolarizations and synaptic events. **c**. Immunocytochemical identification demonstrates the recorded neuron (top panel, filled with Biotin; red) is positive for tyrosine hydroxylase (TH; green). Lower magnification of the immunocytochemical image (lower panel) locates the recorded neuron (arrow) within the ventral region of the VTA ($n = 112$, $N = 80$). **d**. Depiction shows the location of the recorded neuron (large red dot) in relation to all the recorded neurons in this study (gray dots) displayed at three coronal planes. Bottom right panel is a 3D representation of the example cell location (red sphere) in relation to all recorded cells (gray spheres). **e, f** Plot of the mean firing rate (e) and coefficient of variation (CV) for interspike intervals (ISI, f) for each neuron presented in this study. Cyan and Yellow diamonds in panel F highlight the representative cells depicted in Fig. 2a–c and d–f, respectively. **g** Plot of the input resistance for each neuron where resistance was measured. **h** Log-scale plot of the instantaneous firing rate, and its full range, for each neuron presented in this study sorted in the order of increasing median rate. **d–h** The large red symbol in all plots represents the example neuron illustrated in **b** and **c**. Horizontal lines in **e–g** represent median.

Table 2). This comparison to extracellular recordings, and comparisons of the switch from on-cell to whole-cell modes, confirmed that overall firing patterns of dopamine neurons were not affected by the in vivo patch-clamp recording configuration. The range of instantaneous firing rates for each identified dopamine neuron, plotted in Fig. 1h, illustrates the wide range of firing frequencies sampled with in vivo patch-clamp recordings.

Bimodal distributions in subthreshold voltage predict variability of action potential firing pattern.

To best utilize the direct readout of subthreshold membrane potentials provided by in vivo whole-cell recordings, we focused on a systematic investigation of the subthreshold membrane potentials associated with distinct firing patterns. To achieve this with an unbiased approach, we focused on the overall variability of firing (i.e. coefficient of variation (CV)) to characterize the activity of the recorded cell. As shown in Fig. 1f, dopamine neurons displayed a wide spectrum of CVs of ISIs (ISI CV) ranging from 12.9 to 165.7%. Upon inspection of the firing activity of a dopamine neuron with an ISI CV at the low end of this spectrum (12.93%; Fig. 1f, cyan diamond), we found a cell discharging in a single-spike pacemaker pattern (i.e. 0% of spikes fired in bursts (%SFB) according to the 80/160-ms criterion[28]; Fig. 2a–c). To characterize the subthreshold membrane potentials in a comparable fashion, we identified the voltage minima ($V_{min}$) of each ISI, i.e., the most hyperpolarized membrane potential between two action potentials (Fig. 2a, blue circles); as well as the voltage maxima ($V_{thr}$) for each ISI, i.e., the action potential threshold (Fig. 2a, green circles; see methods for action potential threshold definition). The respective distributions of $V_{min}$ and $V_{thr}$ for this example neuron were then fitted with a Gaussian function (Fig. 2b, c, respectively), both of which exhibited a narrow unimodal distribution with low variability ($V_{min}$ CV = 2.7%, $V_{thr}$ CV = 1.9%).

For a dopamine neuron at the high end of the ISI CV spectrum (88.97%; Fig. 1f, blue diamond), the discharge pattern was more irregular and also characterized by transient increases in firing rate that were interspersed by longer silent periods (Fig. 2d–f). During these non-spiking periods, the membrane potential was more hyperpolarized compared to bursting periods (Fig. 2d). The distributions of $V_{min}$ (Fig. 2e) and $V_{thr}$ (Fig. 2f) were also plotted and fitted with Gaussian functions. In contrast to the dopamine neuron from the low end of the ISI CV spectrum, the $V_{min}$ distribution exhibited an overall higher variability ($V_{min}$ CV = 13.4%) and showed a clear bimodal distribution (Fig. 2e). In contrast, the $V_{thr}$ distribution for this cell was unimodal with low variability ($V_{thr}$ CV = 5.6%; Fig. 2f). These two examples of distinct in vivo dopamine neuron firing patterns suggested that the variability of subthreshold membrane potential minima (i.e. $V_{min}$ CV) captured by in vivo patch-clamp recordings might be related to spike train statistics. Indeed, when comparing the $V_{min}$ CV

versus the variability of spike firing pattern (i.e. ISI CV) of the entire dopamine neuron population, a strong positive correlation was detected (Fig. 2g; $V_{min}$ CV vs ISI CV $R^2 = 0.58$; $p = 5.66 \times 10^{-20}$; $n = 97$, $N = 72$). Moreover, $V_{min}$ CV scaled with ISI CV over the entire dynamic range. The variability of the threshold potentials also displayed a clear correlation with spike train regularity (Fig. 2h; $V_{thr}$ CV vs ISI CV $R^2 = 0.32$; $p = 6.65 \times 10^{-10}$; $n = 97$, $N = 72$). Other parameters of subthreshold membrane potentials (mean of $V_{min}$ and $V_{thr}$) and their distributions (skewness and kurtosis of $V_{min}$ and $V_{thr}$) did not correlate well with the ISI CV (Supplemental Fig. 5). However, when we calculated the bimodality coefficients (BC) of $V_{min}$ and $V_{thr}$[34] – a measurement in which higher values are an indication of a bimodal distribution – the resulting values showed a strong, significant correlation with the ISI CV (Fig. 2i; $V_{min}$ BC vs ISI CV $R^2 = 0.29$; $V_{thr}$ BC vs ISI CV $R^2 = 0.09$; $p = 0.001$). In summary, in vivo spike train variability of dopamine neurons was well-correlated, not just with the overall variability of $V_{min}$ and $V_{thr}$, but more specifically with the bimodality coefficient of the $V_{min}$ and $V_{thr}$ distributions. Like the spike train variability itself, the bimodality coefficient was distributed as a spectrum for the recorded dopamine neuron population (Fig. 2i–j).

Pacemaker firing pattern is characterized by unimodal distributions of voltage minima and action potential thresholds.

Based on this analysis, we proceeded to further characterize the subthreshold membrane potentials of low ISI CV (i.e. pacemaker-like) firing patterns of in vivo dopamine neurons, which were now identified by their low BC (lower 10% of the population). Figure 3a, b shows a representative recording and immunocytochemical images of one such neuron located in the VTA. The bottom trace in Fig. 3a illustrates the stability of the in vivo firing pattern when identified by subthreshold membrane potential properties, $V_{min}$ and $V_{thr}$ (blue and green circles, respectively). Similar to the example neuron illustrated in Fig. 2a–c, the $V_{min}$ and $V_{thr}$ distributions of this neuron displayed unimodal distributions (Fig. 3c, d, respectively). Collectively, dopamine neurons with unimodal $V_{min}$ and $V_{thr}$ distributions (i.e. BC < 0.53, see Freeman and Dale[34] for detailed discussion of criteria) had a mean $V_{min}$ BC of $0.29 \pm 0.02$ and mean $V_{thr}$ BC of $0.32 \pm 0.05$ ($n = 10$, $N = 9$). To further examine the relationship between these subthreshold membrane properties and spike firing properties, individual $V_{min}$ and $V_{thr}$ were plotted against their respective ISI durations (Fig. 3e). As the plot shows, this example dopamine neuron with a low $V_{min}$ BC had a narrow range of ISIs, creating a small cluster each for both $V_{min}$ and $V_{thr}$.

To obtain a visual representation of a stereotypical subthreshold membrane potential signature of in vivo pacemaking activity, now defined as a combined profile of unimodal $V_{min}$ and $V_{thr}$ distributions, we utilized event-triggered averaging by superimposing pacemaking spikes (Fig. 3f). The resulting

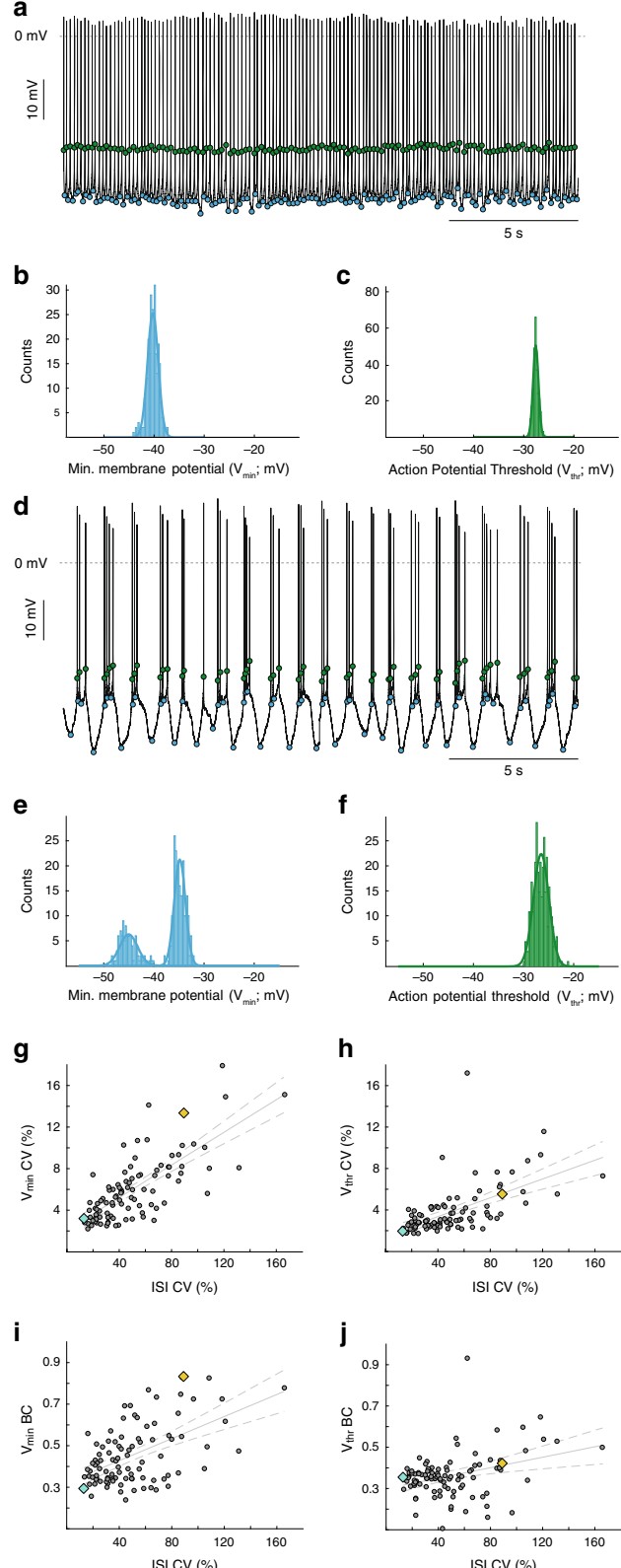

**Fig. 2 Bimodality of distribution for subthreshold voltage predicts variability in action potential firing. a** Example trace from a dopamine neuron firing with a low CV (lower 10%; see Fig. 1f cyan diamond symbol). Blue dots represent the most hyperpolarized membrane potential (minima) reached in between every action potential ($V_{min}$). Green dots represent the most depolarized membrane potential (maxima; action potential threshold) reached in between every action potential ($V_{thr}$). **b**, **c** Distribution of $V_{min}$ (**b**) and $V_{thr}$ (**c**) from the cell in panel **a**, demonstrating unimodal distributions (blue and green lines = Gaussian fits). **d** Example trace from a dopamine neuron firing with a high CV (upper 10%; see Fig. 1f yellow diamond symbol). As in the trace in panel **a**, blue dots represent $V_{min}$ while green dots represent $V_{thr}$ in between spikes. **e**, **f** Distribution of $V_{min}$ and $V_{thr}$ from the cell in panel **d** demonstrating a bimodal (**e**) and unimodal (**f**) distribution, respectively (blue and green lines = Gaussian fits). **g**, **h** Plots correlating variability (measured as CV) of $V_{min}$ (**g**) and $V_{thr}$ (**h**) with variability of ISI (ISI CV) for all cells in this study. **i**, **j** Plots correlating bimodality coefficient (BC) of $V_{min}$ (**i**) and $V_{thr}$ (**j**) with ISI CV for all cells in this study. **g–j** Straight line = linear regression. Dashed lines = 95% confidence interval.

distribution is due to the wide range of pacemaking firing frequencies among all the recorded neurons.

For a more detailed insight, we analyzed the phase plots of action potentials in these pacemaker firing sequences. Figure 3i illustrates an example of a three-action potential sequence (black–orange–red) during pacemaker firing. A phase plot for all spikes occurring during pacemaker firing in this example dopamine neuron is shown in Fig. 3j, with the same three colors (black–orange–red) representing the average of each spike. Note the large degree of overlap throughout the different phases of the action potentials, from threshold to upstroke and overshoot, as well as during repolarization and afterhyperpolarization. Quantitative analysis of all dopamine neurons discharging in this in vivo pacemaker mode identified by their subthreshold properties confirmed a high degree of stability and low variability that extended from the subthreshold domain to the action potential. Indeed, we did not observe significant changes in threshold voltage, maximal depolarization (Max d$V$/d$t$), or repolarization (Min d$V$/d$t$) speeds across the sequence of three action potentials among all low $V_{min}$ BC dopamine neurons (Δ$V_{thr}$: spike 1 vs spike 2 median (IQR) = −0.002 (−0.0043 to 0.0012) mV, spike 2 vs 3 = 0.0016 (−0.0035 to 0.0043) V; Δmax d$V$/d$t$: spike 1 vs 2 = −0.0124 (−0.021 to 0.13) V/s, spike 2 vs 3 = −0.0174 (−0.0387 to 0.002) V/s; $p$ = 0.0375, 0.2324, 0.4316, and 0.02 for all comparisons; $n$ = 10, $N$ = 9; range of 3-spike sequences per cell = 78–691; Fig. 3k and Supplemental Table 2). The low variability of subthreshold membrane potentials combined with stable action potential dynamics demonstrates the capacity of dopamine neurons to maintain their firing stability in vivo even when embedded within active circuits.

**Rebound bursts are characterized by bimodal distribution in voltage minima and hyperpolarizing shift in action potential threshold at the beginning of the burst.**

Next, to further explore the subthreshold membrane potential activity of dopamine neurons on the other end of the spectrum, we studied the dopamine neurons in which subthreshold $V_{min}$ BC was in the top 10% ($V_{min}$ BC range = 0.66–0.83). A representative recording and images of immunocytochemistry of such a dopamine neuron located in the SNc is shown in Fig. 4a, b. Similar to the dopamine neuron described earlier (Fig. 2d–f), the distribution of $V_{min}$ from a bursting dopamine neuron was well-described by a bimodal Gaussian distribution (Fig. 4c, Mode 1 and Mode 2; $V_{min}$ BC = 0.75 ± 0.06; Mode 1 $V_{min}$ mean = −46.3 ± 2.8 mV, standard deviation (SD) = 2.7 ± 1.6 mV; Mode 2 $V_{min}$

averaged trace (Fig. 3f, bold trace) portrays a low-variance subthreshold signature with a ~400-ms gradual, ramp-like membrane potential depolarization of about 10 mV until reaching the action potential threshold. Figure 3g shows the narrow firing frequency distribution of this example cell. The frequency distribution of the population of pacemaker dopamine neurons identified by their low BCs is shown in Fig. 3h. The spread of this

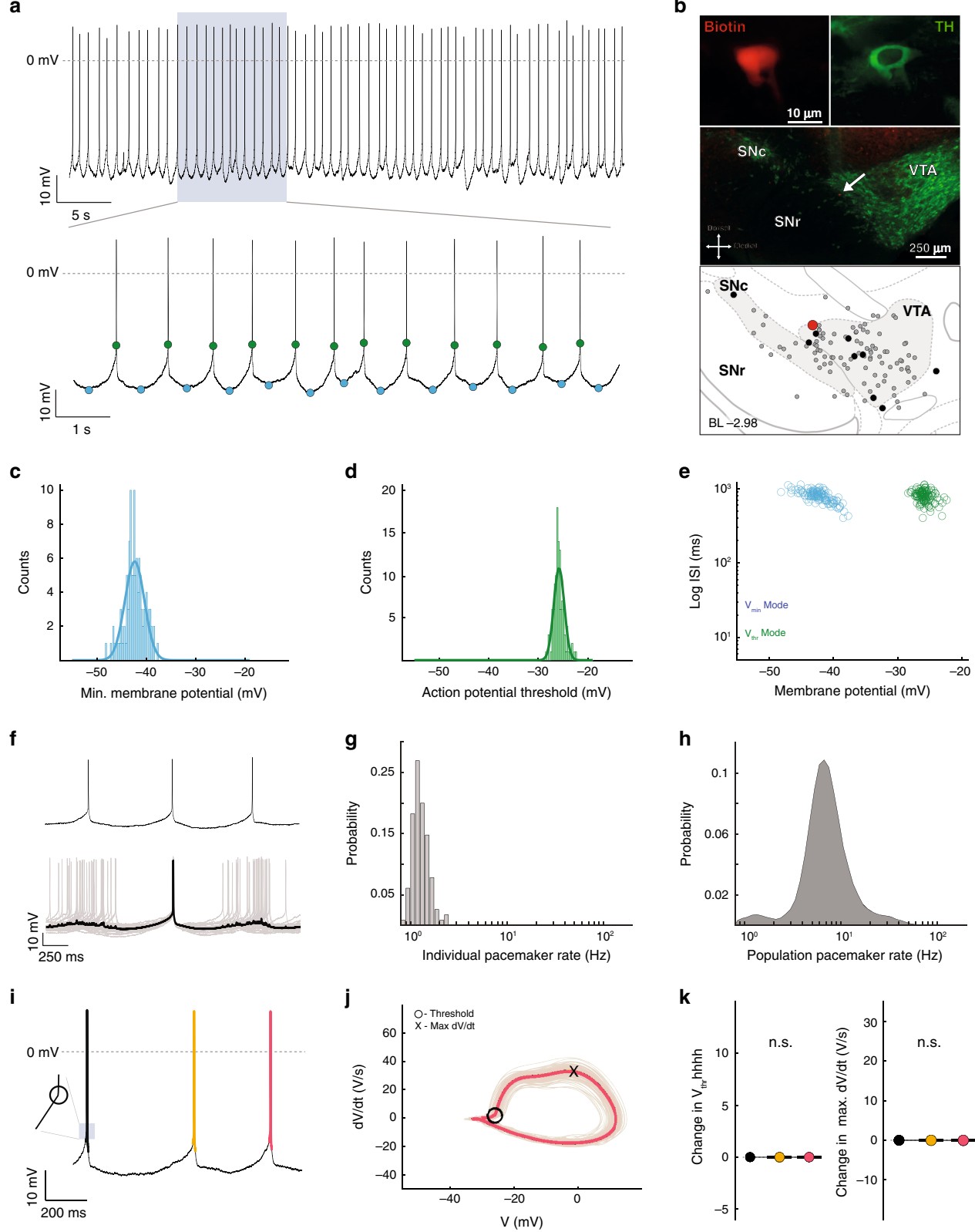

mean $= -37.9 \pm 3.7$ mV, SD $= 1.5 \pm 0.4$; $n = 10$, $N = 10$). As in Fig. 2f, the $V_{thr}$ distribution was also unimodal (Fig. 4d; BC $= 0.38 \pm 0.09$; $V_{thr}$ mean $= -28.91 \pm 2.051$ mV, SD $= 1.73 \pm 0.44$ mV). When plotting the individual ISIs against the corresponding $V_{min}$, the more hyperpolarized membrane potentials (Fig. 4c, Mode 1) were almost exclusively associated with longer ISIs in the range of 600–1200 ms, forming an independent cluster in the plot

(Fig. 4e, dark blue circles), whereas the more depolarized $V_{min}$ were associated with a wide range of shorter ISI durations (Fig. 4e, light blue circles). In contrast, the distribution of $V_{thr}$ in this dopamine neuron had a narrow membrane potential range (Fig. 4e, green circles).

We then applied event-triggered averaging to spikes that immediately followed the Mode 1 $V_{min}$ events, which revealed a

**Fig. 3 Single-spike firing pattern is characterized by a stable and narrow subthreshold membrane potential dynamic range. a** Representative recording of spontaneous in vivo electrical activity from a dopamine neuron in the VTA. The upper trace displays action potentials during a stable recording for >60 s. As in Fig. 2, blue dots in the lower trace represent $V_{min}$ while green dots represent $V_{thr}$. **b** Immunocytochemical identification demonstrates the recorded neuron (red) as TH-positive (green) within the VTA (arrow). Depiction shows the location of the recorded neuron (large red dot) in relation to all the recorded neurons in this study (gray and black dots). Black dots represent the locations of all the neurons that fired with a low $V_{min}$ BC (lower 10%). **c** Distribution of all $V_{min}$ from the cell in **a**, **b**, demonstrating a unimodal distribution (blue line = single Gaussian fit). **d** Distribution of all subthreshold voltage maxima ($V_{thr}$) from the cell in **a**, **b**, demonstrating a unimodal distribution (green line = single Gaussian fit). **e** Scatter plot of the interspike interval (ISI) versus $V_{min}$ (blue) and $V_{thr}$ (green). **f** High temporal resolution of single-spike firing pattern (upper panel), and a mean spike-triggered pacemaker pattern (black trace in lower panel) with overlaid (gray) traces. **g** Histogram of the instantaneous firing rate for the cell in **a**, **b**. **h** Histogram of instantaneous firing rate for all cells with a low $V_{min}$ BC. **i** Example trace from the cell firing in a single-spike pattern with three action potentials in order from black, then orange, then red. Inset illustrates where the threshold occurs for the black action potential. **j** The phase plots for each action potential are displayed in the same color scheme as in **i** (black, orange, and red; plots are nearly completely overlapping). **k** Summary data of the changes in threshold voltage (left) and maximal depolarization rate (Max d$V$/d$t$; right) across the sequence of three action potentials for all cells firing with a low $V_{min}$ BC (two-sided Wilcoxon signed-rank test).

stereotypical, low-variance subthreshold membrane potential signature and associated firing (Fig. 3f). The resulting subthreshold signature was characterized by a curved, long duration (668.5 ± 213.1 ms) trajectory of initial membrane hyperpolarization (10.5 ± 2.7 mV of hyperpolarization to a $V_{min}$ of −47.99 ± 2.5 mV) followed by membrane depolarization back to threshold. The rebound firing occurring after large hyperpolarizations was associated with a transient increase in firing in the beta frequency range. Accordingly, we termed these events *rebound bursts*. In contrast, event-triggered averaging of spikes following the Mode 2 $V_{min}$ events did not result in a stereotypical low-variability subthreshold waveform. The frequency distribution associated with the rebound burst significantly widened the range of dopamine neuron firing (Fig. 4g, h) as compared to pacemaking (pacemaker firing rate SD: 5.5 Hz; rebound burst firing rate SD: 6.9 Hz; F-test $p = 1.064 \times 10^{-30}$), both on the level of individual cells and, in particular, on the level of dopamine neuron populations (compare Figs. 3g and 4g).

The phase plot of the rebound burst action potentials shows different dynamics compared to that of pacemaking (Fig. 4i–k). In particular, the first action potential of the rebound burst (Fig. 4i, orange) had a significantly more hyperpolarized action potential threshold accompanied by an accelerated speed of depolarization (Max d$V$/d$t$; Fig. 4j, k). Once within the rebound burst, the threshold and depolarization speed return close to a pre-burst baseline already with the following spike ($\Delta V_{thr}$: spike 1 vs spike 2 median (IQR) = −3.1 (−5.6 to −2.4) mV, spike 2 vs 3 = 1.0 (0.9–1.1) mV; Δmax d$V$/d$t$: spike 1 vs 2 = 16.3 (13.6–21.9) V/s, spike 2 vs 3 = −5.3 (−8.3 to −3.2) V/s; $p = 0.02$ for all comparisons; $n = 10$, $N = 10$; range of 3-spike sequences per cell = 9–95; Fig. 4k and Supplemental Table 2). Thus, rebound burst dopamine neurons have distinct subthreshold membrane potential signatures with two modes of $V_{min}$, which widen their frequency range, and their observed threshold dynamics occur at burst onset.

Plateau bursts are characterized by bimodal distribution in action potential threshold and depolarizing shift in action potential threshold at the end of the burst.

As apparent from Fig. 2j, some dopamine neurons possessed high BC values for $V_{thr}$. To explore the features of this particular subthreshold membrane potential signature, we analyzed the dopamine neurons in the top 10% of $V_{thr}$ BC. An example trace and its corresponding immunocytochemical images of an identified VTA dopamine neuron are shown in Fig. 5a, b. The Fig. 5a bottom trace illustrates that, in contrast to the rebound burst mode, the action potential threshold of this type of neuron showed a depolarizing shift (light green circles). As expected from this observation, the distribution of $V_{thr}$ was clearly bimodal (Fig. 5d; $V_{thr}$ BC: 0.61 ± 0.13; Mode 1 mean = −33.7 ± 8.9, SD =

1.3 ± 0.5; Mode 2 mean = −27.5 ± 9.8, SD = 2.0 ± 1.1; $n = 10$, $N = 10$), a unique feature that is not observed in pacemaking or rebound bursting. In this particular example, the distribution of $V_{min}$ also displayed a high BC and bimodal distribution (Fig. 1c), although this was not a consistent feature of cells displaying a bimodal distribution in $V_{thr}$ (Fig. 5c; $V_{min}$ BC: 0.59 ± 0.15; Mode 1 mean = −37.7 ± 5.2, SD = 2.7 ± 1.8; Mode 2 mean = −28.2 ± 5.9, SD = 1.2 ± 0.4). When plotting the ISI durations against their respective $V_{thr}$ (Fig. 5e), those $V_{thr}$ residing in the more depolarized distribution mode (Mode 2 in Fig. 5d) were exclusively associated with very short ISI durations in the range of 6–26 ms (Fig. 5e, light green circles). In other words, the bimodal $V_{thr}$ distribution identified high-frequency burst discharges.

Event-triggered averaging of action potentials associated with the $V_{thr}$ Mode 2 distribution revealed a distinct low-variability subthreshold membrane potential signature (Fig. 5f, bottom trace). What emerged was a sharply-rising membrane potential plateau of approximately 6 mV in amplitude and 35 ms in duration with a limited number of 2 to 3 high-frequency, gamma-range spikes riding on top (plateau amplitude median (IQR) = 5.2 (3.7–6.2) mV; plateau duration = 45.8 (17.9–63.3) ms; number of spikes = 2.1 (2.0–2.3); mean maximum spike frequency = 41.4 (27.3–83.4) Hz). As these spikes occurred on a depolarized "plateau" of the membrane potential, we termed this discharge pattern *plateau bursts*. As evident from the average trace (Fig. 5f, bold trace), plateau burst intraburst frequency was less variable compared to intraburst frequencies in rebound bursting (compare Figs. 4f and 5f). Figure 5g, h show the firing frequency distribution of the example cell and the entire population of dopamine neurons identified as discharging in the plateau burst mode, respectively. In contrast to rebound bursting, the firing frequency distribution of plateau bursts is almost exclusively in the gamma range.

When plotting the action potential trajectories of plateau bursts, in contrast to rebound bursts, we found no change in the action potential threshold voltage or waveform properties at the onset of plateau bursts (Fig. 5i–k). This implies that no systematic or predictive threshold dynamics precede the occurrence of plateau bursts in these dopamine neurons, unlike the burst-predictive threshold dynamics of rebound bursts. Or, stated from the neuron's perspective, it was "surprised" by the burst. However, after the onset of plateau bursts, the action potential threshold and dynamics dramatically changed, as subsequent action potentials (in most cases the second action potential within bursts) displayed a large depolarizing shift in the threshold potential accompanied by reduced depolarization speed (Fig. 5j, k), both of which are indications of reduced excitability ($\Delta V_{thr}$: spike 1 vs 2 median (IQR) = 0.39 (−0.35–0.92) mV, $p = 0.19$; spike 2 vs 3 = 4.66 (3.62–4.43) mV, $p = 0.002$; Δmax d$V$/d$t$: spike

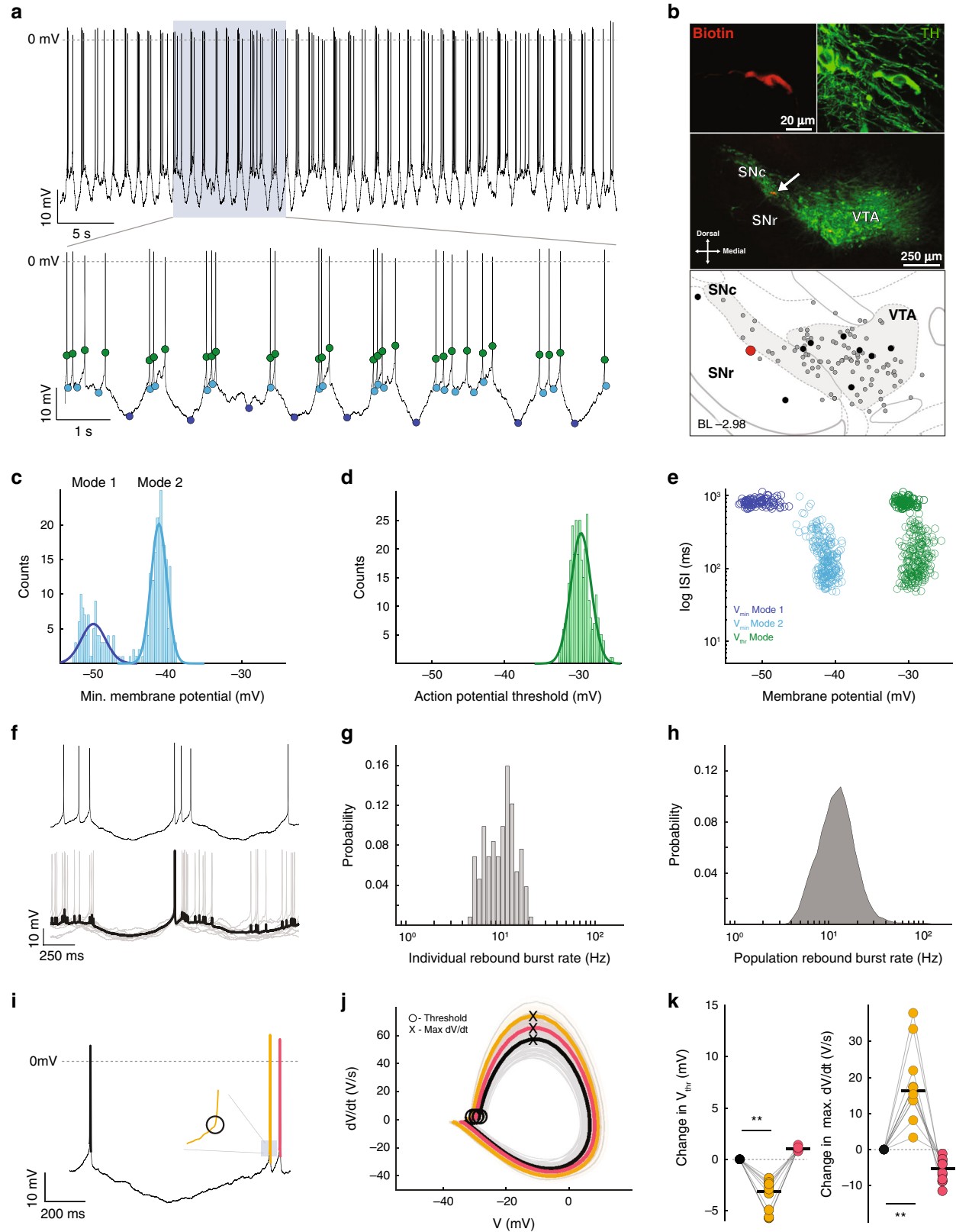

1 vs 2 = −0.5 (−3.25–0.11) V/s, $p$ = 0.11; spike 2 vs 3: −14.38 (−17.95 to −6.73) V/s, $p$ = 0.02, $n$ = 10, $N$ = 10; range of 3-spike sequences per cell = 4–89; Supplemental Table 2). Thus, the threshold dynamics of the plateau burst not only occur in the opposite direction compared to those in rebound bursts, but also served to time the end, not the beginning, of the burst. Indeed, in

over 90 percent of all plateau bursts, the second action potential with the large depolarizing shift in the threshold was also the terminating action potential of the burst (plateau bursts with 2 spikes, $n$ = 455; 3 spikes, $n$ = 48; 4 spikes; $n$ = 11, $N$ = 11). In essence, our unbiased exploration of subthreshold membrane voltages from dopamine neurons in vivo uncovered two unique

**Fig. 4 Rebound burst is characterized by a large hyperpolarization and a hyperpolarized action potential threshold at burst initiation. a** Representative recording of an identified dopamine neuron in the SNc. Both light and dark blue dots in the lower trace represent $V_{min}$ reached in between every action potential. The dark blue dots represent the minima of large hyperpolarizations (Mode 1 in panel **c**). Green dots represent $V_{thr}$. **b** Immunocytochemical identification locates the recorded neuron (arrow) within the VTA. Depiction (bottom panel) shows the location of the recorded neuron (large red dot) in relation to all the recorded neurons in this study (gray and black dots). Black dots represent the locations of all neurons that fired with a high $V_{min}$ BC (top 10%). **c** Distribution of subthreshold voltage minima ($V_{min}$) from the cell in **a**, **b**, demonstrating a bimodal distribution (dark blue line = Mode 1 Gaussian fit; light blue line = Mode 2 Gaussian fit). **d** Distribution of subthreshold voltage maxima ($V_{thr}$) from the cell in **a**, **b**, demonstrating a unimodal distribution (green line = single Gaussian fit). **e** Scatter plot of ISI versus $V_{min}$ and $V_{thr}$. The dark blue cluster is associated with more hyperpolarized voltage membrane potentials and longer duration ISIs. **f** High temporal resolution of a rebound burst (upper panel), and event-triggered rebound burst average (lower panel, black trace) and overlaid (gray) traces. **g** Histogram of the cell firing rate associated with rebound bursts. **h** Histogram of firing associated with rebound bursts for all cells that fired with a high $V_{min}$ BC. **i** Example trace of a cell firing a rebound burst with three action potentials in order from black prior to the burst, then orange beginning the burst, and red. Inset illustrates where the hyperpolarized threshold occurs for the action potential beginning the burst (orange). **j** The phase plots for each action potential are displayed in the same color scheme as in **i**. **k** Summary data of the changes in action potential threshold voltage (left) with significant hyperpolarization in threshold of the first action potential of the burst (orange; $p = 0.02$), and increase in maximal depolarization rate (Max d$V$/d$t$; right; $p = 0.02$; two-sided Wilcoxon signed-rank test) across the sequence of three action potentials for all cells firing in a rebound burst pattern.

subthreshold burst signatures featuring opposing threshold dynamics, each associated with a distinct and non-overlapping frequency range.

Dopamine neurons in vivo fire both Plateau and Rebound bursts during a single recording.

As highlighted in Fig. 2, subthreshold membrane potential features – like spike train statistics – were best described as a spectrum. Indeed, many of the recorded dopamine neurons in the intermediate BC range (10–90%) fired both single spikes and occasional burst-like events (i.e. "mixed mode"). In addition, a handful of dopamine neurons fired in both plateau and rebound bursts as well as in single spikes (Supplemental Fig. 6). A representative trace and immunocytochemical images of an identified VTA dopamine neuron that fired in single spikes with occasional burst events are shown in Fig. 6a, b, respectively. Figure 6a bottom trace illustrates the detail of single spikes interspersed with hyperpolarizing events leading to rebound bursts. This mode of firing was associated with wider unimodal distribution in the $V_{min}$ and $V_{thr}$ (Fig. 6c, d, respectively), sometimes with prominent tails on either or both sides of the distributions; in this case on the left side of the $V_{min}$ distribution. When plotting individual ISIs against their respective $V_{min}$ and $V_{thr}$ (Fig. 6e), wider clustering was observed with no clear segregation. Therefore, we identified events within spike trains by taking advantage of the two dimensions in the scatter plots (Fig. 6e, see Methods section for details). In this example, outlier detection for both ISI (ordinate) and membrane potential (abscissa) values isolated events with hyperpolarized $V_{min}$ and long ISI durations (Fig. 6e, dark blue circles), but not for $V_{thr}$ (Fig. 6e, green circles). Based on the detected events, the event-triggered averaging revealed a subthreshold membrane potential signature similar to the one observed for rebound bursting (Fig. 6f, bottom trace), indicating that the mechanism of burst generation might be similar between dopamine neurons with continuous bursting and those displaying a more "mixed" firing mode. Figure 6g, h show the firing frequency distribution of the example cell and the entire population of dopamine neurons identified as discharging in a mixed pattern, respectively. Note that event-triggered averaging captured fewer high frequency events compared to continuously bursting cells.

In this dopamine neuron firing in a mixed pattern, the phase plot of action potentials in rebound bursts shows similar properties compared to that of dopamine neurons locked in a rebound burst mode (Fig. 6i–k). Again, the first action potential of the rebound burst (Fig. 6i, orange) had a significantly more hyperpolarized action potential threshold coupled with an accelerated speed of depolarization (Max d$V$/d$t$; Fig. 6j, k). Once within the rebound burst, the threshold as well as depolarization speed quickly returned to baseline with the subsequent spike ($\Delta V_{thr}$:

spike 1 vs spike 2 median (IQR) = −2.3 (−2.6 to −1.8) mV, spike 2 vs 3 = 0.6 (0.5–0.9) mV; $\Delta$max d$V$/d$t$: spike 1 vs 2 = 7.18 (6.04–8.98) V/s, spike 2 vs 3 = −1.91 (−2.79 to −1.13) V/s; $p = 4.88 \times 10^{-4}$ for all comparisons; $n = 12$, $N = 12$; range of 3-spike sequences per cell = 1–15; Fig. 6k; and Supplemental Table 2 for cells firing a mixed pattern containing plateau bursts). Thus, the observed threshold dynamics of rebound and plateau bursts in dopamine neurons firing in a mixed pattern are similar to rebound and plateau bursts from dopamine neurons firing in a locked burst mode. Indeed, many of the intrinsic properties were similar among neurons firing in different patterns (Supplemental Fig. 7). The results indicate that firing activity of most dopamine neurons is not well represented by a single state, but is constantly changing in the in vivo environment.

In summary, the first systematic in vivo patch-clamp study of identified midbrain dopamine neurons demonstrated the presence of two qualitatively distinct burst firing mechanisms, identified by their bimodal subthreshold distributions, with contrasting subthreshold membrane potential signatures and diametrically-opposing action potential threshold dynamics. These findings present the first framework of how predictive coding might be biophysically implemented in midbrain dopamine neurons.

## Discussion

For the first time in nearly four decades, our study provides new intracellular in vivo recordings of identified midbrain dopamine neurons. By establishing deep in vivo whole-cell patch-clamp recordings in anaesthetized mice with no noticeable perturbations on firing activity, our large dataset allowed the first systematic analysis of the subthreshold membrane potential range of dopamine neurons in the intact brain. We show that the overall regularity of action potential discharge, captured by the coefficient of variation (CV) of the spike train, correlated well with the CV of the minima ($V_{min}$) and maxima ($V_{thr}$) in the subthreshold membrane potential range. In addition, we demonstrate that bimodality in the distributions of $V_{min}$ and $V_{thr}$ enabled the identification of unique subthreshold signatures associated with distinct burst patterns. We defined (1) a prolonged membrane potential hyperpolarization leading to rebound bursts, which were associated with a hyperpolarizing shift in action potential threshold at the onset of the burst; and (2) a fast depolarization leading to plateau bursts, which were associated with a depolarizing shift in the action potential threshold at the end of the burst. While plateau bursts were strongly coupled to short, high-frequency (gamma range) discharge, rebound bursts occurred at lower frequencies (beta range), some of which would not have been detected by traditional heuristics.

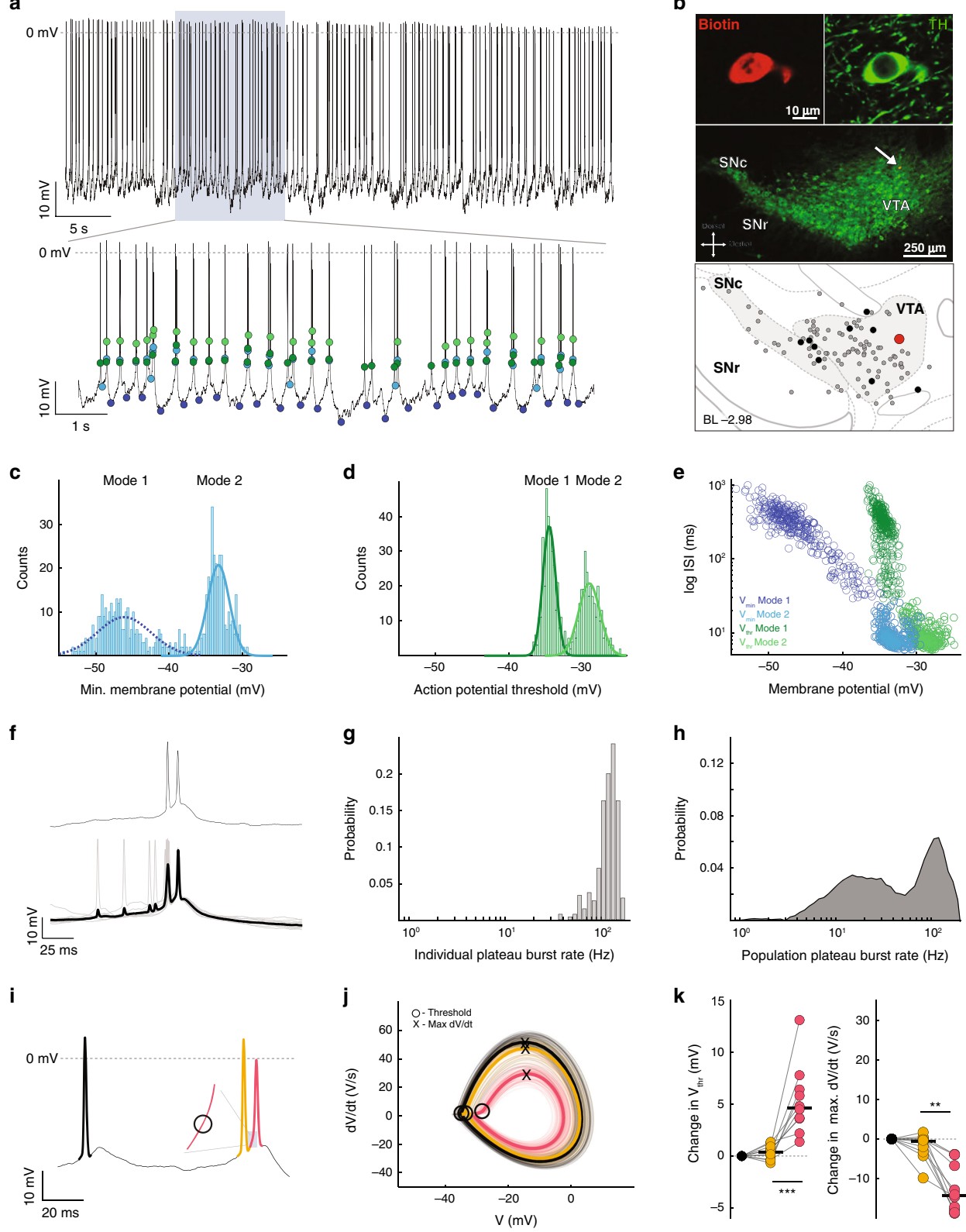

Both plateau and rebound bursts substantially widen the dynamic range of dopamine neuron firing, providing attractive candidate mechanisms that enable predictive coding. The hyperpolarizing shift of the action potential threshold induced in rebound bursts also enhances the potential excitability of the neuron. At the same time, these dopamine neurons remain flexible to allow integration of incoming synaptic inputs.

Moreover, the action potential thresholds of rebound burst spikes are distinct from those of the background single spikes in that they would enable the cell to compute a new prediction for spike timing. Future in vivo whole-cell studies in awake behaving animals will be necessary to explicitly test whether these candidate subthreshold mechanisms are indeed at work during ongoing reward prediction error computations.

**Fig. 5 Plateau burst is characterized by a transient depolarization and a depolarizing action potential threshold at burst termination. a** Representative recording of a dopamine neuron in the VTA. As in Fig. 4, dark blue dots represent more hyperpolarized $V_{min}$ whereas light blue dots represent minima occurring during plateaus. Dark green dots represent $V_{thr}$. Light green dots represent the more depolarized $V_{thr}$. **b** Immunocytochemical identification locates the recorded neuron (arrow) within the VTA. Depiction (bottom panel) shows the location of the recorded neuron (large red dot) in relation to all the recorded neurons in this study (gray and black dots). Black dots represent neurons that fired with a high $V_{thr}$ BC (top 10%). **c** Distribution of all $V_{min}$ from the cell in **a**, **b**, demonstrating a bimodal distribution (dark blue line = Mode 1 Gaussian fit; light blue line = Mode 2 Gaussian fit). **d** Distribution of all $V_{thr}$ from the cell in **a**, **b**, demonstrating a bimodal distribution (dark green line = Mode 1 Gaussian fit; light green line = Mode 2 Gaussian fit). **e** Scatter plot of ISI versus $V_{min}$ and $V_{thr}$. The light green cluster is associated with more depolarized voltage membrane potentials and shorter duration ISIs. **f** High temporal resolution of a plateau burst (upper panel), and event-triggered plateau burst average (black trace in lower panel) with overlaid (gray) traces. **g** Probability histogram of the firing rate associated with plateau bursts for the example cell in **a**, **b**. **h** Probability histogram of firing rate associated with plateau bursts for all cells. **i** Example trace of a cell firing a plateau burst with three action potentials in order from black prior to the burst, then orange beginning the burst, and finally red ending the burst. Inset illustrates where the depolarized threshold occurs for the action potential ending the burst (red). **j** Phase plots for each action potential are displayed in the same color scheme as in **i**. **k** Summary data of the changes in threshold voltage (left) with significant depolarization in threshold of the action potential ending the burst (red; $p = 0.002$); and significant decrease in maximal depolarization rate (Max $dV/dt$; right) of the last action potential (red; $p = 0.02$; two-sided Wilcoxon signed-rank test) for all cells firing in a plateau burst pattern.

In contrast, the observed threshold dynamics of the short, high-frequency plateau bursts might primarily serve temporal precision of firing by controlling the termination of high-frequency bursts. These features render the plateau burst mechanism a plausible candidate for coding unexpected and salient sensory events. While the subthreshold signature of the rebound burst suggests a timed series of initial inhibition followed by excitation, short plateau bursts might be generated by rapid fluctuations in the excitation/inhibition (E/I) balance. Indeed, our previous dynamic clamp studies in vitro revealed similar threshold and action potential waveform behavior when the E/I balance rapidly shifts toward more excitation or disinhibition[35]. In essence, our study identified in vivo operating principles of discharging dopamine neurons and provides explicit candidate mechanisms, which can now be tested in awake animals.

While the in vivo intracellular recordings with sharp micro-electrodes of identified dopamine neurons from the 1980s did not provide systematic analyses of the subthreshold membrane potential range in relation to firing patterns, those results are mostly in accordance with our findings. There is, however, one important exception. Input resistances in our in vivo patch-clamp recordings of dopamine neurons are about 10-fold higher than those reported previously, indicating that the in vivo "high conductance" state might have been overestimated and, in part, be caused by impaling dopamine neurons with sharp microelectrodes[30,31]. In addition, it was only now possible to determine that the in vivo intracellular recordings did not alter the firing pattern of intact cells – both by on-cell recordings in this study and by comparison to previous in vivo extracellular recordings of identified dopamine neurons[19,36]. Importantly, all dopamine neurons included in our study were filled, recovered, and identified as TH-immunopositive. Given these quality control measures and the size of the dataset, we can begin to also address the in vivo subthreshold diversity of dopamine neurons. The previously reported differences in in vitro intrinsic subthreshold properties[18,20,37–39], such as sag amplitudes and rebound latencies, were also evident in our in vivo dataset, although dopamine neurons in the paranigral aspect of the VTA were under-represented. Also, our present in vivo patch-clamp study does not provide information about axonal projections of the recorded dopamine neurons. However, the observed diversity might contribute to subpopulation-specific computations, as also recently identified by in vivo GCaMP imaging during a complex behavioral task[40].

Despite the strengths, our study has a limitation in that we recorded under anaesthesia and were therefore unable to study subthreshold membrane potential signatures of dopamine neuron firing patterns in response to sensory cues or reward-related signals. However, the stability of brain states in controlled anaesthesia is also an advantage as it enabled the unbiased identification of subthreshold membrane potential signatures. Another limitation of our study is that we did not go beyond the identification of subthreshold patterns to explicitly target underlying synaptic and intrinsic biophysical mechanisms. For instance, the deep hyperpolarization that precedes rebound bursting could be activated by different types of synaptic input involving distinct receptors ranging from GABA_A, GABA_B, dopamine receptor D2 (D2R), or metabotropic glutamate receptors (mGluR) coupled to small conductance calcium-activated potassium (SK) channels[35,41–46]. In addition, the rebound kinetics are likely to be controlled by intrinsic conductances in the subthreshold range. Here, rebound depolarization accelerating (e.g. HCN channels[38]) and delaying (e.g. Kv4 channels[39,47]) conductances are likely to be relevant. Some dopamine neurons even respond to the termination of a hyperpolarizing current injection with a robust recruitment of T-type calcium channels[37,48]. The plateau burst, on the other hand, is driven by a rapid onset depolarization, which is likely to be caused by a rapid change in the E/I balance[44]. Fundamentally, rebound and plateau bursts represent responses within dopamine neurons comprised of a sequence of synaptic inputs that are active at any moment in the intact brain in addition to the intrinsic cellular processes that are particular to the individual neuron. Future studies exploring detailed intrinsic biophysical properties of in vivo dopamine neurons in the context of synaptic input will be necessary to determine the degree to which synaptic inputs contribute to rebound and plateau bursts in dopamine neurons. In addition, dynamic clamp and pharmacological manipulations could be performed to further characterize intrinsic activity underlying these burst patterns. In essence, in vivo patch-clamp studies will facilitate future mechanistic studies of dopamine neuron firing in anaesthetized as well as awake behaving animals.

## Methods

**Animals.** C57BL/6N (Charles River Laboratories and Janvier Labs) and C57BL/6J (Jackson Laboratories) mice (67 males and 7 females, aged 8–16 weeks) were used for the experiments. Mice were maintained on a 12-h light/dark cycle with food and water available ad libitum. All experimental procedures were approved by the German Regional Council of Darmstadt (TVA 54-19c20/15-F40/28) or the University of Texas at San Antonio Institutional Animal Care and Use Committees.

**Stereotactic surgeries.** For head-plate implantations, animals were anaesthetized with isoflurane (1.0–2.5% in 100% O2, 0.35 L/min) and placed in a stereotaxic frame (David Kopf Instruments). The body temperature was maintained at 37–38 °C with a heating blanket, and the breathing rate at 1–2 Hz. Small indentations were made above the VTA (AP: −3.08 mm, ML: ± 0.25–0.75 mm) or SNc (AP: −3.08 mm, ML: 0.9–1.4 mm) as a reference for later craniotomies.

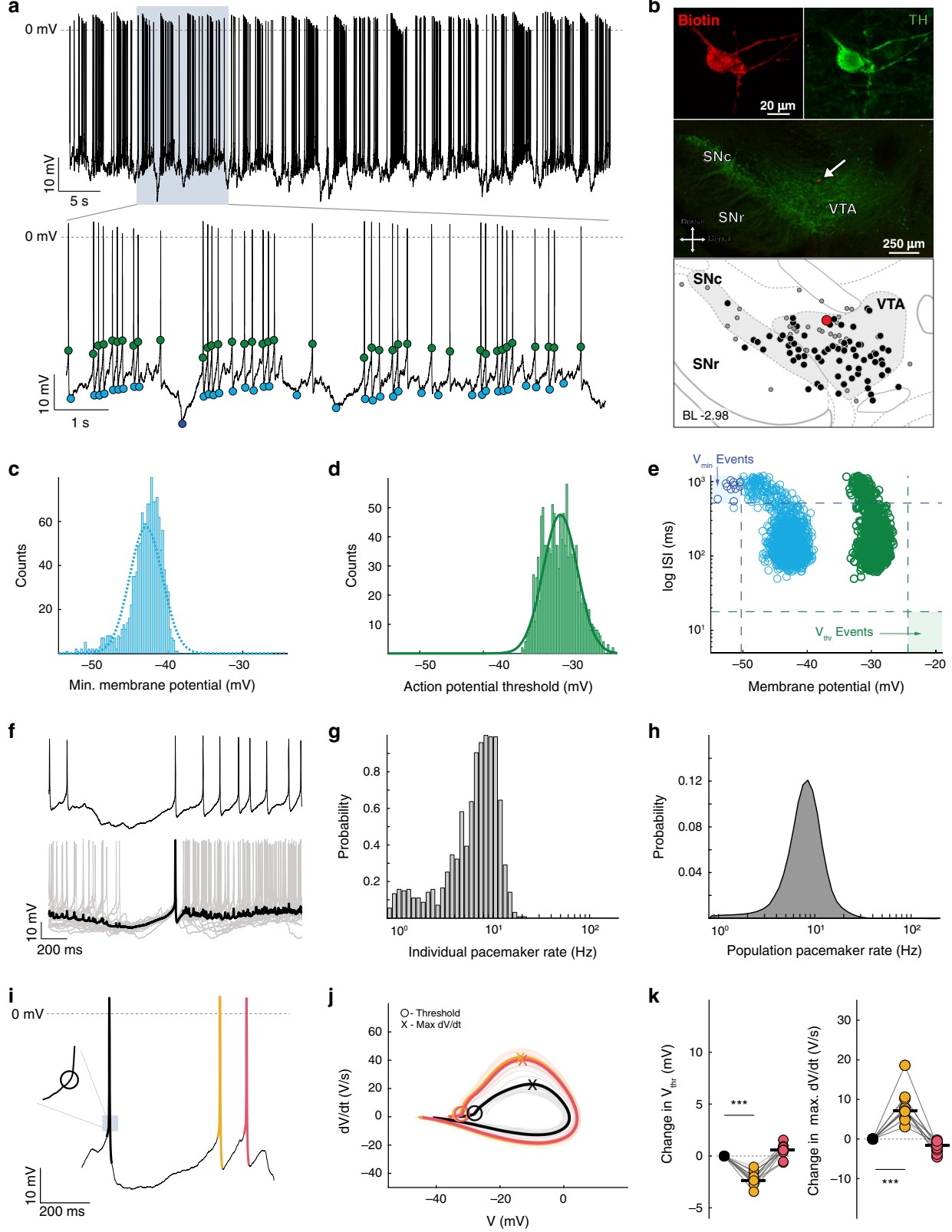

Customized head-plates were anchored to the skull with superglue and dental cement (Paladur, Kulzer), as well as with screws mounted on the skull.

**In vivo electrophysiology**. Electrophysiology was performed using glass electrodes (5–10 MΩ) filled with internal solution containing (in mM): 135 K-gluconate, 10 HEPES, 5 KCl, 5 MgCl₂, 0.1 EGTA, 0.075 CaCl₂ (variant 1) or no added CaCl₂ (variant 2), 5 ATP (Na), and 1 GTP (Li), neurobiotin (0.1%; Vector Laboratories)

or biocytin (0.3%), at pH 7.35. The estimated free calcium concentrations based on a calcium contamination of 15 μm (for details see Woehler et al.[49]) were about 80 nM for variant 1 and ~40 nM for variant 2 (variant 1: $n = 104$, $N = 74$: variant 2: $n = 8$, $N = 6$). Recording electrodes were made with borosilicate glass capillaries (G120F-4, Warner Instruments) pulled with a pipette puller (DMZ Universal Electrode Puller, Zeitz-Instruments).

Animals were maintained under isoflurane (1.0–2.5% in 100% O₂, 0.35 L/min) and head-fixed to a customized recording platform (Luigs & Neumann). The body

**Fig. 6 Rebound bursts occur in cells firing in a mixed pattern including bursts and single spikes. a** Representative recording of an identified dopamine neuron in the VTA. Blue dots in the lower trace represent $V_{min}$. The dark blue dot represents $V_{min}$ during a large hyperpolarization. Green dots represent $V_{thr}$. **b** Immunocytochemical identification (top panel) demonstrates the recorded neuron (red) as TH-positive (green) within the VTA. The depiction (bottom panel) shows the location of the recorded neuron (large red dot) in relation to all the recorded neurons (gray and black dots). Black dots represent the locations of all neurons that fired in an intermediate $V_{min}$ BC (10–90%). **c, d** Distribution of $V_{min}$ (**c**) and $V_{thr}$ (**d**) from the cell in **a**, **b**. **e** Scatter Plot of ISI versus $V_{min}$ and $V_{thr}$. Blue dashed lines indicate outlier identification in the upper left plane (dark blue circles) for $V_{min}$. Green dashed lines indicate outlier identification in the lower left plane for $V_{thr}$ with no events detected. **f** High temporal resolution of a rebound burst (upper panel), and event-triggered rebound burst average (lower panel, black trace) and overlaid (gray) traces. **g** Histogram of the cell firing rate associated with rebound bursts. **h** Histogram of firing associated with rebound bursts for all cells that fired in a mixed pattern. **i** Example trace of the cell firing a rebound burst with three action potentials in order from black prior to the burst, then orange beginning the burst, and red. Inset illustrates where the action potential threshold occurs prior to the burst (black). **j** The phase plots for each action potential are displayed in the same color scheme as in **i**. **k** Summary data of the changes in action potential threshold voltage (left) with significant hyperpolarization in threshold of the first action potential of the burst (orange; $p = 4.88 \times 10^{-4}$), and change in maximal depolarization rate (Max d$V$/d$t$; middle; $p = 4.88 \times 10^{-4}$; two-sided Wilcoxon signed-rank test) with significant increase in Max d$V$/d$t$, across the sequence of three action potentials for all rebound bursts from cells firing in a mixed pattern.

temperature (37–38 °C) and breathing rate (1-2 Hz) were continuously monitored. Craniotomies and duratomies were performed at the reference locations. The electrode was lowered down to ~200 μm above the region of interest with a positive pressure of 600–1000 mbar. The pressure was gradually lowered and kept in a range between 40 and 70 mbar during probing for cells in the VTA (DV: 4.0–5.2 mm) and SNc (DV: 3.8–5.2 mm). A tone generator (PSA-12, HEKA; Custom script, Axograph) was used to monitor the pipette resistance during probing and sealing. A fluctuating 20–50% increase in the pipette resistance indicated a cell was nearby. Once reaching proximity, pressure was released to achieve a GΩ seal. On-cell recordings were obtained for some neurons prior to breaking in. After breaking into the whole-cell mode with a small suction, cell capacitance and series resistance were estimated. Whole-cell recordings of spontaneous activity were recorded in current-clamp mode with zero holding current ($I = 0$) with a series resistance range of 20–130 MΩ–using long taper (10 mm) pipettes to reach the ventral midbrain – for a median duration of approximately 5 min (median = 4 min 52 s; range = 50 s–69 min 37 s) including spontaneous activity and current-injection protocols. Responses to hyperpolarizing current injections were then obtained in current-clamp mode to measure sag and rebound delay. Recording signals were obtained with a patch-clamp amplifier (EPC10 USB, HEKA; Multiclamp 700B, Molecular Devices) and the data were acquired with PatchMaster software (HEKA) or Axograph software (Axograph) at a sampling rate of 20 kHz.

**Histology and microscopy**. After recording, animals were injected with a lethal dose of pentobarbital (0.3–0.4 mL) and transcardially perfused with PBS followed by 4% paraformaldehyde (wt/vol) and 15% picric acid (vol/vol) in phosphate-buffered saline. Collected brains were post-fixed in 4% paraformaldehyde and 15% picric acid overnight and then stored in PBS at 4 °C. Coronal slices were collected at 50-μm-thick using a vibratome. Collected slices were washed with PBS and then incubated in blocking solution (0.01 M PBS, 10% horse serum (vol/vol), 0.5% Triton X-100 (vol/vol), 0.2% BSA (vol/vol)) for 2 h at room temperature. For the tyrosine hydroxylase (TH) staining, polyclonal rabbit antibody (catalog no. 657012, Merck, 1:500) was used as the primary antibody diluted in carrier solution (0.01 M PBS, 1% horse serum, 0.5% Triton X-100, 0.2% BSA) overnight at 4 °C on a shaker. Slices were then rinsed 3 times in 1X PBS and incubated for 2 h in AlexaFluor-488 goat anti-rabbit IgG (catalog no. A11008, ThermoFisher, 1:1000) diluted in the carrier solution previously described as the secondary antibody. Neurobiotin and biocytin were visualized with AlexaFluor-568 streptavidin conjugate (catalog no. S11226, ThermoFisher, 1:1000). Sections were washed 5x in PBS and then mounted on slides (Vectashield, Vector) using Prolong Gold Antifade Reagent (Fisher, Cat# 9071S) mounting media, and visualized under a light microscope (BX53, Olympus) and a laser-scanning microscope (LSM510 Meta, Zeitz).

**Anatomical localization of labeled neurons**. Imaged brain sections were registered to the Allen Mouse Brain Common Coordinate Framework (CCF) using an automated section-to-atlas registration in Neuroinfo software (MBF Bioscience). Registration of imaged sections results in a transform that has assigned image pixels to brain regions in the Allen Reference Atlas (Allen Institute for Brain Science). The location of neurobiotin- or biocytin-labeled neurons are marked within the registered sections and cell locations are transformed into CCF. The resulting transforms of cell locations provides exact coordinates for each recorded cell within a 3-dimensional reference brain.

**Spike detection**. A spike peak was detected if the following three conditions were met within a 3-ms window after crossing a d$V$/d$t$ threshold (default value = 10 mV/ms): First, the minimal value of d$V$/d$t$ must be <−5 mV/ms. Second, the maximal value of voltage must be within 30 mV of the highest voltage in the entire trace. Last, the difference between the maximal voltage and the voltage at the d$V$/d$t$-threshold crossing (spike height) must be >5 mV. In some cases, these parameters were modified to improve spike detection performance.

Detection of spike threshold was dependent on the d$V$/d$t$ and d$^2V$/d$t^2$ crossing certain thresholds. To make the algorithm flexible and robust, these thresholds were indexed to the maximal values of the first and second voltage derivative for each detected spike. Specifically, the spike threshold is detected when the following conditions are met within a 2-ms window ending at the spike peak: Both first and second derivative of voltage cross 5% of their respective within-window maximal values for two consecutive samples. This ensured that spike threshold determination was adjusted for each spike. The phase of the spike being detected as threshold was visually verified and was found to be very consistent. Similar to the above algorithm, the numeric parameters were tunable.

**Identification of rebound and plateau bursts**. Rebound bursts: a Gaussian mixture model with two Gaussians was fitted to the $V_{min}$ distribution of neurons with the highest (top 10%) BC values. The voltage midway between the crossing point of the two individual Gaussians and the peak of the leftward Gaussian was chosen as the threshold for classifying $V_{min}$ values as deep hyperpolarizations. If either of the two ISIs following a deep hyperpolarization were faster than the cell's median firing rate, they were identified as a rebound burst. The burst was said to be terminated when the instantaneous firing rate fell below the median. In non-bimodal cells, deep hyperpolarizations showed up as outliers in the ISI vs $V_{min}$ scatter plot. As such, we used Tukey's method to detect outlier ISI and $V_{min}$ values. If a $V_{min}$ was more hyperpolarized than q1-2.5x (q3-q1), where q1 = 25$^{th}$ quantile and q3 = 75$^{th}$ quantile in the $V_{min}$ distribution; and if the corresponding ISI was longer than q3 + 2.5x (q3-q1), where q1 = 25$^{th}$ quantile and q3 = 75$^{th}$ quantile in the ISI distribution, it was classified as a deep hyperpolarization. Identification of the subsequent burst followed the same criteria as for bimodal cells. The factor 2.5 was derived from analysis of bimodal cells and to optimize detection in non-bimodal cells.

Plateau bursts: For dopamine neurons with the highest (top 10%) $V_{thr}$ BC, the voltage criterion for classifying $V_{thr}$ values as plateau potentials was calculated in a similar way to deep hyperpolarizations, with the only difference being that it was calculated on $V_{thr}$ distributions and the second Gaussian in the $V_{thr}$ distribution lay to the right of the main Gaussian. All ISIs whose $V_{thr}$ values were higher than this criterion were considered part of the plateau burst. In non-bimodal cells, plateau potentials were detected in a parallel fashion to deep hyperpolarizations, with 2.5 used as the factor for $V_{thr}$ outlier detection and 0.5 as the factor for ISI outlier detection, in addition to the $V_{thr}$ being 3.5 mV more depolarized than its immediate predecessor for it to be identified as a plateau burst. Also, the corresponding ISI has to be shorter than q1-0.5x (q3-q1), where q1 = 25$^{th}$ quantile and q3 = 75$^{th}$ quantile in the ISI distribution. Similar to rebound burst detection, these criteria were derived from bimodal cells and with an aim to optimize detection in non-bimodal cells.

**Quantification and statistical analysis**. Data processing and quantifications were performed using MATLAB (MathWorks). Wilcoxon rank-sum tests were used to determine statistical differences using MATLAB (MathWorks). Graphs and maps were created using MATLAB or Prism 6. Bimodality coefficient was used for dual process detection[34]. Measurements of effect size were also calculated[50]. Parametric data were presented as mean ± standard deviation. Non-parametric data were presented as median with interquartile range. Wilcoxon rank-sum test and Wilcoxon sign rank tests were used for unpaired and paired data. Adjusted R squared values are reported for linear correlation.

**Reporting summary**. Further information on research design is available in the Nature Research Reporting Summary linked to this article.

## Data availability
The data used to generate the results are available at a Harvard Dataverse repository at https://dataverse.harvard.edu/dataset.xhtml?persistentId=doi:10.7910/DVN/Q9ZXLZ. Source data are provided with this paper.

## Code availability

The custom code used for analyzing this data, along with sample data and documentation, has been uploaded to a Github repository at https://github.com/anandsku/DA_invivo.

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

## Acknowledgements

We are grateful to Nao Uchida, who helped define the relevant in vivo patch-clamp recording parameters of dopamine neurons. We thank Kauê M. Costa for his help with confocal imaging and initial mapping of dopamine neurons. We are grateful for excellent technical support by Jasmine Sonntag and Beatrice Fischer (Neurophysiology, Goethe University Frankfurt). We thank Gerard Beaudoin for custom scripting of Axograph tone generation. We thank Stephan Lammel (UC Berkeley) for his advice on the manuscript, and Lora Kovacheva on the design of the figures. This work was funded by grants to J.R. (DFG CRC1080, DFG CRC1193, and NIH DA041705) and C.A.P. (NIH MH113341, MH107229).

## Author contributions

J.R. and C.A.P. designed the study and established the technique for in vivo patch-clamp recordings of identified dopamine neurons during C.A.P.'s sabbatical at Goethe

University, Frankfurt. K.O., J.P. and C.A.P. performed the experiments. Analysis was carried out by A.K., S.S., K.O., and J.P. supervised by J.R. and C.A.P.

## Funding

## Competing interests
The authors declare no competing interests.
