## [Peer Review File · Nature Communications]

REVIEWER COMMENTS

Reviewer #1 (Remarks to the Author):

This paper primarily describes a valuable new dataset of intracellular recordings from dopamine neurons (DA) in anesthetized mice. Foundational work in the field from Grace and Bunney provided key insights into the physiology of DA that led to the ability to identify DA in vivo extracellular recordings in behaving animals. This was a critical step that underlies substantial future progress discovering the role of DA in reinforcement learning and distinguishing those roles from previously postulated functions. This subfield has advanced substantially and now a number of additional questions have been raised. The most pressing are those that surround understanding the diversity of DA, and mechanisms that underlie the generation of the fascinating neural correlates of DA. Much of the debate around these questions has turned on neural correlates of behavior - does DA activity reflect salient sensory input, errors in value estimates, initiation of movement, components of motor commands per se, aversive signaling, sensory responses, novelty, etc. The authors here argue, persuasively in my opinion, that knowledge of the underlying biophysics of DA will be a critical part of this phase of research into DA function. This large dataset of whole cell patch clamp recordings in vivo will provide an important advance.

My main comments are relatively straightforward and do not require additional experimentation.

First, I found that the paper seemed to equivocate on a key point that made it difficult to connect to existing debates in the field. Specifically, do the authors believe their data is most consistent with the claim that known biophysical differences between DA subtypes can be revealed by the analyses of spontaneous activity observed in this dataset or are those biophysical differences 'overwhelmed' by differences in input activity that tend to drive the observed bursting patterns? Quite a bit of the paper sounds like the former, but I think the impact of the paper will be helped by clarifying this distinction and the authors' current interpretation of the data in this regard.

Some of the evidence I find in the paper that is difficult to reconcile with the former (i.e. intrinsic biophysics determine in vivo activity):

1. It was unclear from Supp Fig 2 whether there were clear anatomical gradients in observed physiology (otherwise I think they would have been described) as has been argued from in vitro work in the Roper lab. I think it is ok if this is not borne out in these data (there may be many explanations) but I believe the paper would be better with some discussion of this point. For example, why wasn't ML position used as used previously in Lammel 2008? Also curious why the sag magnitude or delay to spike properties were not correlated with bursting properties, peak firing rate, etc. I was interested in a more quantitative comparison than "The previously reported differences in in vitro intrinsic subthreshold properties 18, 20, 41-43, such as sag amplitudes and rebound latencies, were also evident in our in vivo dataset, although dopamine neurons in the paranigral aspect of the VTA were underrepresented."
2. At one point the manuscript indicates that individual DA neurons could be observed to exhibit multiple types of bursting patterns indicating that it may be more a consequence of afferent input than intrinsic biophysics.
3. The authors note that DA cells in vivo have, surprisingly, high input impedance. This quite different than classic measurements. They assert " This high in vivo input resistance indicates that differences in intrinsic excitability of dopamine neurons are less likely to be shunted in comparison to other types of central neurons". Shouldn't a high input resistance allow afferent input to produce large fluctuations in membrane potential and thus substantially alter spiking? I think this needs to be clarified, otherwise it would seem to provide an argument that afferent input can be a relatively dominant driver of changes in spiking activity.

Second, the core distinction that the paper focuses on - the two "diametrically opposed" patterns of bursting - is very difficult to understand in the text of the paper. I think some additional work

clarifying the description here would be really helpful to a broad audience that may not be accustomed to some of the language. One example is that the following sentence in the abstract is really long and difficult to parse as a reader:

We demonstrate that dopamine neuron activity in vivo deviates from a single-spike pacemaker pattern to transient increases in firing rate via two diametrically opposing biophysical mechanisms:

1. a transient depolarization resulting in high frequency plateau bursts associated with a reactive, depolarizing shift in action potential threshold;
2. and a prolonged hyperpolarization preceding slower rebound bursts characterized by a predictive, hyperpolarizing shift in action potential threshold.

I had to copy it over and edit a bit to try and follow that sentence.

Similarly, later in the paper I thought this sentence was tough to understand:

"We demonstrate that pacemaking dopamine neurons in vivo have the ability to tightly control their excitability, as monitored by their action potential thresholds and interspike membrane potentials, and that opposing mechanisms underlying burst discharge are characterized by predictive and reactive shifts in action potential thresholds, and baseline membrane potentials."

Third, it would be quite interesting if the authors did any stimulation of neurons with current injection or simulated input in vivo. I guess since it is not reported the authors did not do this, but I think it might be useful to mention as a goal for future experiments. I understand the awake, behaving experiments are interesting, but for exploring intrinsic biophysics substantial insight can be gained from direct manipulation via current injection.

Fourth, the paper indicates that the method developed here will be influential. The description of the recordings in the methods however is pretty slim. It would be helpful to have additional methodological description and some additional details about recording quality - duration of recordings, series resistance, etc.

Reviewer #2 (Remarks to the Author):

The manuscript "Subthreshold repertoire and threshold dynamics of midbrain dopamine neuron firing in vivo" presents novel and technically outstanding study examining the subthreshold membrane potential dynamics during action potential firing in dopaminergic neurons using in vivo whole cell patch clamp. To my knowledge, this study presents the first whole-cell recordings of midbrain dopaminergic neurons from mice in the in vivo preparation. The authors find that dopamine neurons fire two types of bursts: 1) transient depolarization-induced bursting or 2) hyperpolarization-enabled rebound bursting. This distinction has not yet been made experimentally because most studies use extracellular spiking recordings in vivo which do not allow direct measurements of subthreshold membrane potential. Therefore, the data here features subthreshold membrane potential measurements that could not have been obtained using any other method. This manuscript presents an important advance and should be published in Nature Communications. I congratulate the authors on this heroic effort.

Comments:

1. An important question is whether the depolarization-induced plateau bursts result from synaptic input. Using existing data (if at the appropriate voltages to minimize IPSPs, near Erev GABA), the authors should quantify (or comment on) the frequency of putative EPSPs in plateau bursting cells. The authors may also consider comparing/commenting on synaptic membrane polarizations or membrane noise in plateau bursting cells vs rebound bursters and single spikers.

2. In addition to excitatory synaptic input or tone, one possibility is that the plateau bursting cells are

more depolarized on average due to their intrinsic membrane properties. Please report and compare a) the average non-spiking membrane potential and b) the input resistance for each population of cells reported here (single spikers, plateau bursters and rebound bursters).

3. In the plot of dV/dt vs V_m in Fig 3J, there seems to be a prominent axon initial segment component in the dV/dt plot as compared to the plots shown in Fig 2J and 4J. Did rebound bursting cells always exhibit a stronger AIS mediated component of the dV/dt plot?

4. Please provide a clear plot of the absolute numbers and the fraction of cells that were categorized as single spiker, rebound burster and plateau bursters throughout the entire population recorded. Along those lines, do single spikers bursters comprise distinct populations or is there a continuum? Plotting how each individual cell met these criteria would make the categorization of all cells transparent for the reader (biomodality coefficient for min V_m and APthresh or CV).

5. The authors state that many of their cells recorded were "locked in burst modes." Were 'hybrid' neurons observed (eg. single spiking cells that switch to bursting) and how many? Also on Pg 14, the authors mention cells that fire both plateau and rebound patterns. How many exhibited this behavior? Were these VTA and/or SNc cells?

6. How does spike threshold for action potentials rebounding from 'natural' hyperpolarization compare to the action potential threshold for the first spike when the cell is recovering from somatic hyperpolarization (during steps used to test I_h , for example)? This would yield more information about the intrinsic vs synaptic mechanisms of the AP threshold change during rebound bursting.

7. The findings here should be discussed in the context of a recent study from one of the coauthors that examines spontaneous bursting using juxtacellular recordings (Farassat et al 2019). Although axon projections are not available here, the authors can comment on whether the activity of neurons in lateral and medial SNc and in VTA shown here roughly matches with what was previously reported.

Minor comments:

- Please report the number of cells recorded in the two different internal solutions, the 40nM and 80 nM solution. Also, the methods on page 19-20 describe variant 1 solution with 0.075 mM added Ca as having 40 nM free Ca and variant 2 with no added calcium as having 80 nM free Ca. Should this be reversed?
- Please provide more thorough description of how action potential threshold was calculated.
- The assumption is that the traces here represent spontaneous activity recorded with zero holding current. If true, please state this clearly in the methods.
- On Pg. 14, a reference was made to 'a small spikelet at the end of the plateau burst.' Currently it indicates is 'data not shown' but simply showing a trace could be informative.
- In general, references to 'data not shown' should be minimized. I would encourage the authors to show these data.
- How input resistances measured here in vivo compare to values reported in brain slices should be added to the discussion of this topic in the paper.
- Color coding rebound and plateau bursters as well as single spikers in Panels E and G of supplemental figure 2 would be informative for the reader.
- Please report the range of how many 3 spike sequences were included per cell in data points shown in Figures 2K, 3K, and 4K.
- On Pg 7, the authors state "This high in vivo input resistance indicates that differences in intrinsic excitability of dopamine neurons are less likely to be shunted in comparison to other types of central neurons." This statement is unclear.
- On Pg. 11, the authors comment that their data are 'in line with ideas of distributed population coding in dopamine neurons.' This comment should moved to the Discussion where a full explanation of this point can be provided.

Reviewer #3 (Remarks to the Author):

In this study the researchers investigate the sub threshold physiological dynamics that underly midbrain dopamine neuron firing in vivo with whole cell recording. This has been a long standing question in the field in particular because ex vivo recordings generally show quite different firing patterns from those reported with extracellular recording in vivo, leaving a distinct gap in understanding of how in vivo firing (which tends to appear more stochastic and include burst activity) is generated. The authors identify three firing modes across the population of confirmed dopaminergic neurons and compare various physiological characteristics of these firing patterns. Overall, the study is well designed and systematic, and likely to be of interest to anyone interested in dopamine neuron dynamics. Some points should be addressed:

1. The authors describe "burst" firing patterns of the neurons, and do not seem to identify burst events using the Grace and Bunney characteristics. While this is sensible, no information on the method used to classify activity as "bursting" is provided. Is it simply based on the underlying membrane potential? Please include more specific information on this in the methods. Also, most readers of this paper are going to assume that the "bursting" here is classified by the Grace and Bunney parameters, so a discussion about how these different firing patterns are similar to or deviate from that pattern would be informative to readers.
2. The firing pattern identified here as "rebound burst firing" looks very similar to what has been termed in other brain regions as "up and down states". Especially given that this firing pattern lacks the very short ISIs seen in the bursting illustrated in Figure 4, what is the rationale or heuristic for identifying this as burst firing? How is it different from up/down states? For instance, the "bursts" in Figure 3 do not show a decreasing AP spike height across subsequent APs within a burst, a property common to "bursting" and evident in Figure 4. Should this impact the classification of this firing pattern as "bursting"?
3. In Figure 4, in the lower trace in panel A, some of the ISIs are so brief it is impossible to tell in this x axis scale in some cases if there are 1, 2, or 3 APs, or a spikelet that was not classified as an AP. More resolution on the x scale would be informative for understanding the Vm dynamics and AP ISI durations in this example.
4. Related to this, how was an event classified as a non-AP "spikelet" rather than a full AP?
5. Given that a variety of differences between SNc and VTA neurons' membrane conductances, including those that support pacemaker firing and burst induction, have been reported, were any further differences between SNc and VTA neural firing observed, beyond that high frequency bursting neurons were only detected the VTA?
6. Minor point: in Figure 1, the distribution of the firing rates looks bimodal - is this correct? Is the input resistance distribution also bimodal? Do these distributions correlate with anything, for instance neuron location?
7. Minor point: because of the way the last sentence in the abstract is phrased, it seems implied that the two bursting firing patterns identified here are concluded to separately be responsible for "sensory cue and prediction error coding in dopamine neurons", which is not actually what is demonstrated by this dataset.

We thank the editors and reviewers for providing constructive peer review, which helped make the manuscript more accurate, clearer, and better contextualized. A common theme among all three reviewers was to supply more detailed information on the data collected. To this end, we have provided further analysis to the results section and added 2 new main figures (new Figures 2 and 6) and new supplemental figures. Perhaps a greater common theme was to clarify whether dopamine neuron firing patterns are separate categories or opposite ends of a spectrum. We show, particularly with the new Figures 2 and 6, and Supplemental Figure 6, that dopamine neurons actually are part of a continuum of firing patterns with a mix of rebound bursts, plateau bursts, and single spike firing. We have made significant changes throughout the text to remove any potential confusion, and more clearly state that *in vivo* firing pattern and subthreshold signatures are in a spectrum. Responses to specific comments are provided below with the reviewer comments copied in *italicized text*.

Reviewer 1 comments

This paper primarily describes a valuable new dataset of intracellular recordings from dopamine neurons (DA) in anesthetized mice. Foundational work in the field from Grace and Bunney provided key insights into the physiology of DA that led to the ability to identify DA *in vivo* extracellular recordings in behaving animals. This was a critical step that underlies substantial future progress discovering the role of DA in reinforcement learning and distinguishing those roles from previously postulated functions. This subfield has advanced substantially and now a number of additional questions have been raised. The most pressing are those that surround understanding the diversity of DA, and mechanisms that underlie the generation of the fascinating neural correlates of DA. Much of the debate around these questions has turned on neural correlates of behavior - does DA activity reflect salient sensory input, errors in value estimates, initiation of movement, components of motor commands per se, aversive signaling, sensory responses, novelty, etc. The authors here argue, persuasively in my opinion, that knowledge of the underlying biophysics of DA will be a critical part of this phase of research into DA function. This large dataset of whole cell patch clamp recordings *in vivo* will provide an important advance.

My main comments are relatively straightforward and do not require additional experimentation.

1. *“First, I found that the paper seemed to equivocate on a key point that made it difficult to connect to existing debates in the field. Specifically, do the authors believe their data is most consistent with the claim that known biophysical differences between DA subtypes can be revealed by the analyses of spontaneous activity observed in this dataset or are those biophysical differences ‘overwhelmed’ by differences in input activity that tend to drive the observed bursting patterns? Quite a bit of the paper sounds like the former, but I think the impact of the paper will be helped by clarifying this distinction and the authors’ current interpretation of the data in this regard.”*

Since dopamine neurons *in vitro* generally only fire in a pacemaker pattern, we find that the *in vivo* firing pattern of dopamine neurons, as revealed by their subthreshold activity, likely results from a combination of intrinsic properties and synaptic input. We did not find any large correlations in the degree of bursting and intrinsic properties. Indeed, the majority of dopamine neurons *in vivo* fired in a mixed pattern with combinations of bursts and single spike firing. We have clarified this point in the last paragraph of the discussion.

2. *"It was unclear from Supp Fig 2 whether there were clear anatomical gradients in observed physiology (otherwise I think they would have been described) as has been argued from in vitro work in the Roeper lab. I think it is ok if this is not borne out in these data (there may be many explanations) but I believe the paper would be better with some discussion of this point. For example, why wasn't ML position used as used previously in Lammel 2008?"*

The new Supplemental Figure 4 has now been updated by the addition of comparisons that did not turn out to be significant. In general, we did not find any correlation in sag amplitude or rebound delay in the medial-lateral direction. This point has been added to the discussion.

3. *"Also curious why the sag magnitude or delay to spike properties were not correlated with bursting properties, peak firing rate, etc. I was interested in a more quantitative comparison than "The previously reported differences in in vitro intrinsic subthreshold properties 18, 20, 41-43, such as sag amplitudes and rebound latencies, were also evident in our in vivo dataset, although dopamine neurons in the paranigral aspect of the VTA were underrepresented."*

We have now added correlations on a number of measurements (input resistance, sag, rebound delay, etc.) with cell location (VTA vs SNC in Supplemental Figure 3; medial lateral location in Supplemental Figure 4; rebound or plateau bursts in Supplemental Figure 7). These comparisons were not originally reported due to their lack of correlation. However, we agree with the reviewers (see also reviewer 2 comment 9) that this information will be of value to readers.

4. *"At one point the manuscript indicates that individual DA neurons could be observed to exhibit multiple types of bursting patterns indicating that it may be more a consequence of afferent input than intrinsic biophysics."*

We have now added a more comprehensive description of all the number of cells that display rebound bursts and plateau bursts. We also included the frequency of plateau and rebound burst occurrences. These have been added to new Figure 6 and Supplemental Figure 6 (See also reviewer 2 comment 6). We agree with the reviewer that a significant driver of bursting is afferent input.

5. *"The authors note that DA cells in vivo have, surprisingly, high input impedance. This quite different than classic measurements. They assert " This high in vivo input resistance indicates that differences in intrinsic excitability of dopamine neurons are less likely to be shunted in comparison to other types of central neurons". Shouldn't a high input resistance allow afferent input to produce large fluctuations in*

membrane potential and thus substantially alter spiking? I think this needs to be clarified, otherwise it would seem to provide an argument that afferent input can be a relatively dominant driver of changes in spiking activity.”

We have added a clearer sentence, “This high *in vivo* input resistance indicates that the intrinsic excitability of dopamine neurons is less likely to be shunted in comparison to other types of central neurons.” (See reviewer 2 comment 18)

6. *“Second, the core distinction that the paper focuses on - the two “diametrically opposed” patterns of bursting - is very difficult to understand in the text of the paper. I think some additional work clarifying the description here would be really helpful to a broad audience that may not be accustomed to some of the language.”*

The term has been replaced in the abstract to “qualitatively distinct”. “Diametrically opposing” has been kept later in the manuscript after clarification about the opposing mechanisms of plateau and rebound bursts (e.g. depolarization vs. hyperpolarization, respectively).

7. *“Similarly, later in the paper I thought this sentence was tough to understand: “We demonstrate that pacemaking dopamine neurons *in vivo* have the ability to tightly control their excitability, as monitored by their action potential thresholds and interspike membrane potentials, and that opposing mechanisms underlying burst discharge are characterized by predictive and reactive shifts in action potential thresholds, and baseline membrane potentials.”*

The sentence has been re-written to “Moreover, we demonstrate that *in vivo* pacemaking dopamine neurons tightly control their action potential threshold, while rebound and plateau bursts are characterized by opposing shifts in action potential thresholds.”

8. *“Third, it would be quite interesting if the authors did any stimulation of neurons with current injection or simulated input *in vivo*. I guess since it is not reported the authors did not do this, but I think it might be useful to mention as a goal for future experiments. I understand the awake, behaving experiments are interesting, but for exploring intrinsic biophysics substantial insight can be gained from direct manipulation via current injection.”*

We agree. Future experiments using these stimulations are planned. This has been added to the last paragraph of the discussion section.

9. *“Fourth, the paper indicates that the method developed here will be influential. The description of the recordings in the methods however is pretty slim. It would be helpful to have additional methodological description and some additional details about recording quality - duration of recordings, series resistance, etc.”*

The methods section now has a more comprehensive description on details of the recordings. We have added sections for burst detection, subthreshold event detection, action potential threshold definition, and added more detail to methods that were already in the first submission of the manuscript. The recording durations ranged from about 1 to 70 minutes. Series resistances ranged from 20 – 130 M Ω , using long taper (10 mm) pipettes to reach the ventral midbrain.

Reviewer 2 comments

The manuscript “Subthreshold repertoire and threshold dynamics of midbrain dopamine neuron firing in vivo” presents novel and technically outstanding study examining the subthreshold membrane potential dynamics during action potential firing in dopaminergic neurons using in vivo whole cell patch clamp. To my knowledge, this study presents the first whole-cell recordings of midbrain dopaminergic neurons from mice in the in vivo preparation. The authors find that dopamine neurons fire two types of bursts: 1) transient depolarization-induced bursting or 2) hyperpolarization-enabled rebound bursting. This distinction has not yet been made experimentally because most studies use extracellular spiking recordings in vivo which do not allow direct measurements of subthreshold membrane potential. Therefore, the data here features subthreshold membrane potential measurements that could not have been obtained using any other method. This manuscript presents an important advance and should be published in Nature Communications. I congratulate the authors on this heroic effort.

Comments:

1. *“An important question is whether the depolarization-induced plateau bursts result from synaptic input. Using existing data (if at the appropriate voltages to minimize IPSPs, near Erev GABA), the authors should quantify (or comment on) the frequency of putative EPSPs in plateau bursting cells. The authors may also consider comparing/commenting on synaptic membrane polarizations or membrane noise in plateau bursting cells vs rebound bursters and single spikers.”*

We have commented on possible sources for plateau and rebound bursts in the discussion. While membrane fluctuation analysis proved difficult due to action potential-mediated conductances, we find that the silent and “quiet” dopamine neurons appear to be dominated by GABA_A receptor mediated currents (reversal of fluctuations is at the chloride reversal potential at -60 mV; see Supplemental Figure 1).

2. *“In addition to excitatory synaptic input or tone, one possibility is that the plateau bursting cells are more depolarized on average due to their intrinsic membrane properties. Please report and compare a) the average non-spiking membrane potential and b) the input resistance for each population of cells reported here (single spikers, plateau bursters and rebound bursters).”*

We added membrane potential, input resistance, for all cells in this study in the new Supplemental Figure 7. In general, we find no large differences among the different firing patterns. Since the firing pattern of dopamine neurons *in vivo* occurs as a continuum, we compared the various measurements by the frequency of events occurring in a recording.

3. *“In the plot of dV/dt vs V_m in Fig 3J, there seems to be a prominent axon initial segment component in the dV/dt plot as compared to the plots shown in Fig 2J and*

4J. *Did rebound bursting cells always exhibit a stronger AIS mediated component of the dV/dt plot?*

All firing pattern events (single spike, rebound burst, plateau burst) had cells containing action potentials with both prominent and non-prominent axon initial segment components. The example phase plots were chosen without regard to the presence of the AIS component due to the lack of any tendency for any firing pattern to have one.

4. *“Please provide a clear plot of the absolute numbers and the fraction of cells that were categorized as single spiker, rebound burster and plateau bursters throughout the entire population recorded.”*

Since we make it clearer now that the firing pattern of dopamine neurons *in vivo* occurs in a continuum, cells are no longer categorized. Instead, to highlight the extremes of the continuum of firing patterns, we selected the dopamine neurons with the top and lowest 10% bimodality coefficient for V_{\min} or V_{thr} . As such, the example cells are meant to highlight the extreme ends of the spectrum, with the cells in the 10-90% range having a mixed firing pattern, sometimes with both plateau and rebound bursts during a single recording (see Supplemental Figure 6). Therefore, we do not provide absolute numbers of categories.

5. *“Along those lines, do single spikers bursters comprise distinct populations or is there a continuum? Plotting how each individual cell met these criteria of would make the categorization of all cells transparent for the reader (bimodality coefficient for $\min V_m$ and AP_{thresh} or CV).”*

There is a continuum (see comment 6, next). We added a plot of bimodality coefficient versus ISI coefficient of variation to the new Figure 2 and new Supplemental Figure 5, showing that the firing pattern of dopamine neurons correlates highly with bimodality coefficient for V_{\min} and V_{thr} , but not for other distribution measures such as kurtosis or skewness. Many cells were found to show a mix of firing patterns that included single-spike firing interspersed with rebound or plateau bursts.

6. *“The authors state that many of their cells recorded were “locked in burst modes.” Were ‘hybrid’ neurons observed (eg. single spiking cells that switch to bursting) and how many?”*

We have now added a more comprehensive description of all the number of cells that display rebound bursts, plateau bursts, and cells firing in a mixed pattern (e.g single spike firing with plateau bursts). Indeed, dopamine neuron firing pattern, as determined by subthreshold criteria, ranges in a spectrum with single spiking at one end, and rebound and plateau bursts at the other end. We included a new Figure 6 and Supplemental Figure 6 illustrating the dopamine neurons firing in a mixed pattern (See reviewer 1 comment 4).

7. *“Also on Pg 14, the authors mention cells that fire both plateau and rebound patterns. How many exhibited this behavior? Were these VTA and/or SNc cells?”*

We have now added a more comprehensive description of all the number of cells that display rebound bursts and plateau bursts (see new Supplemental Figure 6). A comparison of the occurrences of plateau and rebound bursts in SNc vs VTA in new Supplemental Figure 3. We found 4 dopamine neurons that displayed both rebound and plateau bursts during a single recording.

8. *“How does spike threshold for action potentials rebounding from ‘natural’ hyperpolarization compare to the action potential threshold for the first spike when the cell is recovering from somatic hyperpolarization (during steps used to test I_h , for example)? This would yield more information about the intrinsic vs synaptic mechanisms of the AP threshold change during rebound bursting.”*

We found no difference in spike threshold dynamics resulting from large spontaneous hyperpolarizations or from hyperpolarizations elicited by current injection, with means of 3.28 and 3.29 mV shifts, respectively.

9. *“The findings here should be discussed in the context of a recent study from one of the coauthors that examines spontaneous bursting using juxtacellular recordings (Farassat et al 2019). Although axon projections are not available here, the authors can comment on whether the activity of neurons in lateral and medial SNc and in VTA shown here roughly matches with what was previously reported.”*

We have now added correlations on a number of measurements with cell location (VTA vs SNC, medial lateral location, etc.). These comparisons were not originally reported due to their lack of correlation. However, we agree with the reviewers (see also reviewer 1 comment 3) that this information will nonetheless be of value to readers, and has been added to new Supplemental Figures 3 and 4.

10. *“Please report the number of cells recorded in the two different internal solutions, the 40nM and 80 nM solution. Also, the methods on page 19-20 describe variant 1 solution with 0.075 mM added Ca as having 40 nM free Ca and variant 2 with no added calcium as having 80 nM free Ca. Should this be reversed?”*

Of the dopamine neurons that fired spontaneously, the number of cells recorded in 40 nM Ca^{++} was 8 cells from 6 mice. The number of cells recorded in 80 nM Ca^{++} was 104 cells from 74 mice. These values and others have been added to the results section in the correct order.

11. *“Please provide more thorough description of how action potential threshold was calculated.”*

The detection of spike threshold was dependent on the dV/dt and d^2V/dt^2 crossing certain thresholds specific for each neuron. A detailed description has been added to the methods section in the subsection regarding spike detection.

12. *“The assumption is that the traces here represent spontaneous activity recorded with zero holding current. If true, please state this clearly in the methods.”*

The reviewer is correct. Spontaneous activity was recorded with zero current injection ($I = 0$). This clarification has been added to the methods section.

13. "On Pg. 14, a reference was made to 'a small spikelet at the end of the plateau burst.' Currently it indicates is 'data not shown' but simply showing a trace could be informative."

A spikelet is simply a small amplitude action potential, which indicates the development of depolarization block due to sodium channel inactivation. Since this is simply a small amplitude spike at the end of a burst, the term "spikelet" has been removed to eliminate confusion. (See reviewer 3 comment 6)

14. "In general, references to 'data not shown' should be minimized. I would encourage the authors to show these data."

All references to "data not shown" have been replaced except for one. The only remaining "data not shown" refers to the recordings of identified TH-negative neurons. All the other previous occurrences of "data not shown" are now replaced with new data in the results section, new main figures, and new supplemental figures.

15. "How input resistances measured here *in vivo* compare to values reported in brain slices should be added to the discussion of this topic in the paper."

We agree. We have added a quantitative comparison of *in vivo* and *in vitro* input resistances. We find that dopamine neurons *in vivo* have approximately 1-2 nS higher conductance compared to recordings *in vitro*.

16. "Color coding rebound and plateau bursters as well as single spikers in Panels E and G of supplemental figure 2 would be informative for the reader."

The original color coding and specific labeling has been kept in new Supplemental Figure 4 (previous supplemental figure 2) because additional colors increased the number of colors and symbols to 6. This made the figure too confusing when we tried it. However, readers can gain a sense of the location of the cells firing in different patterns from panel B in Figures 3-6.

17. "Please report the range of how many 3 spike sequences were included per cell in data points shown in Figures 2K, 3K, and 4K."

The ranges have been added to the Methods section. The ranges are as follows:

Pacemaker	78 to 691 triplets
Rebound	9 to 95 triplets
Plateau	4 to 89 triplets
Mixed	1 to 15 triplets

18. "On Pg 7, the authors state "This high *in vivo* input resistance indicates that differences in intrinsic excitability of dopamine neurons are less likely to be shunted in comparison to other types of central neurons." This statement is unclear."

The statement has been changed to "This high *in vivo* input resistance indicates that the intrinsic excitability of dopamine neurons is less likely to be shunted in comparison to other types of central neurons." (See reviewer 1 comment 5)

19. *“On Pg. 11, the authors comment that their data are ‘in line with ideas of distributed population coding in dopamine neurons.’ This comment should be moved to the Discussion where a full explanation of this point can be provided.”*

The comment has been removed.

Reviewer 3 comments

In this study the researchers investigate the sub threshold physiological dynamics that underly midbrain dopamine neuron firing in vivo with whole cell recording. This has been a long standing question in the field in particular because ex vivo recordings generally show quite different firing patterns from those reported with extracellular recording in vivo, leaving a distinct gap in understanding of how in vivo firing (which tends to appear more stochastic and include burst activity) is generated. The authors identify three firing modes across the population of confirmed dopaminergic neurons and compare various physiological characteristics of these firing patterns. Overall, the study is well designed and systematic, and likely to be of interest to anyone interested in dopamine neuron dynamics. Some points should be addressed:

1. *“The authors describe “burst” firing patterns of the neurons, and do not seem to identify burst events using the Grace and Bunney characteristics.”*

We have added a direct comparison of bursting identified by subthreshold signatures in this study, with the bursting identified by the 80/160 ms criterion in mice under similar recording conditions from Farassat et al., 2019. (See reviewer 3 comment 3)

2. *“While this is sensible, no information on the method used to classify activity as “bursting” is provided. Is it simply based on the underlying membrane potential? Please include more specific information on this in the methods.”*

The methods section now contains a description for burst definitions using our subthreshold criteria. In general, we find that bimodality coefficient of subthreshold measures (V_{\min} , V_{thr}) correlate well with the coefficient of variation of interspike intervals (ISI CV). Individual subthreshold events are identified as, e.g., those V_{\min} in the hyperpolarized mode of the bimodal distribution. Upon examining those subthreshold events, we find that the firing rate transiently increases to above the median rate for the cell. In essence, we let the cell membrane potential inform us when “bursting” should occur.

3. *“Also, most readers of this paper are going to assume that the “bursting” here is classified by the Grace and Bunney parameters, so a discussion about how these different firing patterns are similar to or deviate from that pattern would be informative to readers.”*

We agree. We have now added in the results (page 18) a direct comparison of the plateau and rebound bursts in this study, which are determined by subthreshold criteria (i.e. the cell tells us when an event occurs), with the heuristic criteria defined by Grace and Bunney (ISI beginning with ≤ 80 ms and ending with an ISI of ≥ 160

ms) in Farassat et al., 2019. In general, when comparing our subthreshold criteria to the 80/160 ms heuristic directly in our recordings:

For Rebound Bursts:

- 58.43 of rebound bursts meet the 80/160 ms criterion defined by Grace and Bunney
- On a per cell basis, the range of DHPs that meet the 80/160 ms criteria is 0 to 100% with a median of 68.33%.

•

For Plateau Bursts:

- 94.53% of plateau bursts meet the 80/160 ms criterion
- On a per cell basis, the range of plateau bursts that meet 80/160 ms criterion is 0-100% with a median of 98.97%

Out of 2269 bursts in our entire dataset that meet the 80/160 ms criterion:

- 854 have 2 spikes
- Median number of spikes is 3
- Maximum number of spikes is 101
- 36 bursts have > 20 spikes
- 9 bursts have > 50 spikes

4. *“The firing pattern identified here as “rebound burst firing” looks very similar to what has been termed in other brain regions as “up and down states”. Especially given that this firing pattern lacks the very short ISIs seen in the bursting illustrated in Figure 4, what is the rationale or heuristic for identifying this as burst firing? How is it different from up/down states? For instance, the “bursts” in Figure 3 do not show a decreasing AP spike height across subsequent APs within a burst, a property common to “bursting” and evident in Figure 4. Should this impact the classification of this firing pattern as “bursting”?”*

As mentioned in reviewer 3 comment 2, above, individual subthreshold events are identified by their bimodal distribution, or outliers in scatter plots for mixed pattern cells. Upon examining those subthreshold events, if the firing rate transiently increases to above the median rate for the cell it is identified as a burst. A more comprehensive description of burst identification has been added to the Methods section.

5. *“In Figure 4, in the lower trace in panel A, ... it is impossible to tell in this x axis scale in some cases if there are 1, 2, or 3 APs, or a spikelet that was not classified as an AP. More resolution on the x scale would be informative for understanding the Vm dynamics and AP ISI durations in this example.”*

While we agree with the reviewer that individual spikes are difficult to distinguish in that trace, we chose to keep the trace in its original temporal scale to maintain a consistent scale across Figures 3-6. Higher temporal resolution of plateau bursts can still be viewed in panels F and I of Figure 5 (previously figure 4).

6. *“Related to this, how was an event classified as a non-AP “spikelet” rather than a full AP?”*

A spikelet was simply a smaller amplitude action potential. Therefore, the term “spikelet” has been removed to reduce confusion. (see reviewer 2 comment 13)

7. *“Given that a variety of differences between SNc and VTA neurons’ membrane conductances, including those that support pacemaker firing and burst induction, have been reported, were any further differences between SNc and VTA neural firing observed, beyond that high frequency bursting neurons were only detected the VTA?”*

We added a more comprehensive comparison of the VTA and SNc in Supplemental Figure 3. The only additional difference among all the new comparisons was that dopamine neurons in the SNc tend to fire at a slower mean rate than those located in the VTA.

8. *“Minor point: in Figure 1, the distribution of the firing rates looks bimodal - is this correct? Is the input resistance distribution also bimodal? Do these distributions correlate with anything, for instance neuron location?”*

Following the suggestion, we tested for bimodality, but found that the firing rate and input resistance distributions are not bimodal according to the same criteria we use to determine bimodality for the V_{\min} and V_{thr} distributions. Only those neurons with high coefficient of variation of ISI have high bimodality coefficient (Figure 2). Other comparisons are included in Supplemental Figures 4, 5, 7.

9. *“Minor point: because of the way the last sentence in the abstract is phrased, it seems implied that the two bursting firing patterns identified here are concluded to separately be responsible for “sensory cue and prediction error coding in dopamine neurons”, which is not actually what is demonstrated by this dataset.”*

The last sentence in the abstract has been changed to “Our findings define a mechanistic framework for the biophysical implementation of dopamine neuron firing patterns in the intact brain.”

REVIEWERS' COMMENTS

Reviewer #1 (Remarks to the Author):

The authors did an admirable job of adding additional data and responding to all comments. I am fully satisfied with the revision.

Reviewer #2 (Remarks to the Author):

The ordering of the supplemental figures is incorrect and may have been reversed during the upload process. Otherwise, the additional figures and analysis have certainly improved the paper. The authors have addressed all of my concerns.

Reviewer #3 (Remarks to the Author):

The authors have adequately addressed all of my previous comments. A few very minor points remain:

1. I think supplementary Figs 3 and 4 are switched in order between the text and the Captions.
2. For the correlation plots, is the P = Pearson's Correlation Coefficient? Some of the correlation plots, especially in the Supplementary material do not seem to have correlation coefficients reported. Also, the authors may note that while $p < 0.05$ in many of the correlation plots, the variability is great, as detected by low R^2 values.
3. In supplementary figure 2C and E, since the authors have on cell and whole cell data from the same cells for the measurements, it would be most appropriate to use a paired statistical analysis. It is not clear if the tests were paired or unpaired.
4. In supplementary figure 3 ("SNc and VTA dopamine neurons have similar electrophysiological characteristics in vivo"), while clearly the means are not different for most of these measurements, there do appear to be differences in the variance between SNc and VTA. This might be worth noting.

Reviewer #1 (Remarks to the Author):

The authors did an admirable job of adding additional data and responding to all comments. I am fully satisfied with the revision.

We thank the reviewer.

Reviewer #2 (Remarks to the Author):

The ordering of the supplemental figures is incorrect and may have been reversed during the upload process. Otherwise, the additional figures and analysis have certainly improved the paper. The authors have addressed all of my concerns.

The Supplemental Figures 3 and 4 figure captions were reversed and are now in the correct order (See Reviewer 3 comment 1).

Reviewer #3 (Remarks to the Author):

The authors have adequately addressed all of my previous comments. A few very minor points remain:

1. I think supplementary Figs 3 and 4 are switched in order between the text and the Captions.

The Supplemental Figures 3 and 4 figure captions were reversed and are now in the correct order (See Reviewer 2).

2. For the correlation plots, is the P = Pearson's Correlation Coefficient? Some of the correlation plots, especially in the Supplementary material do not seem to have correlation coefficients reported. Also, the authors may note that while $p < 0.05$ in many of the correlation plots, the variability is great, as detected by low R squared values.

The R -squared values are the adjusted R -squared as described in the Methods. The p -values reported are computed p -values for Pearson's correlation using a Student's t distribution for a transformation of the correlation. This is a standard measure that adjusts for degrees of freedom in the data and makes R -squared comparable across data sets.

3. In supplementary figure 2C and E, since the authors have on cell and whole cell data from the same cells for the measurements, it would be most appropriate to use a paired statistical analysis. It is not clear if the tests were paired or unpaired.

The analysis was done with paired statistical testing. This has been clarified in the figure legend.

4. In supplementary figure 3 ("SNc and VTA dopamine neurons have similar electrophysiological characteristics in vivo"), while clearly the means are not different for most of these measurements, there do appear to be differences in the variance between SNc and VTA. This might be worth noting.

Upon checking for variances we found no differences between VTA and SNc.